# Transcriptional rewiring of an evolutionarily conserved circadian clock

Alejandra Goity[1,2], Andrey Dovzhenok[3], Sookkyung Lim[3], Christian Hong[4], Jennifer Loros[5,6], Jay C Dunlap [5] & Luis F Larrondo [1,2✉]

## Abstract

Circadian clocks temporally coordinate daily organismal biology over the 24-h cycle. Their molecular design, preserved between fungi and animals, is based on a core-oscillator composed of a one-step transcriptional-translational-negative-feedback-loop (TTFL). To test whether this evolutionarily conserved TTFL architecture is the only plausible way for achieving a functional circadian clock, we adopted a transcriptional rewiring approach, artificially co-opting regulators of the circadian output pathways into the core-oscillator. Herein we describe one of these semi-synthetic clocks which maintains all basic circadian features but, notably, it also exhibits new attributes such as a "lights-on timer" logic, where clock phase is fixed at the end of the night. Our findings indicate that fundamental circadian properties such as period, phase and temperature compensation are differentially regulated by transcriptional and posttranslational aspects of the clockworks.

**Keywords** Circadian Rhythms; Transcriptional Rewiring; Neurospora; Synthetic Biology; Photoresponses
**Subject Categories** Biotechnology & Synthetic Biology; Chromatin, Transcription & Genomics

## Introduction

Circadian rhythms are present in diverse organisms, from bacteria to mammals. These ~24-h rhythms, generated by circadian oscillators, share common features: they are maintained under constant conditions, are temperature compensated (period is stable within a physiological range of temperatures) and are entrained by environmental cues such as light and temperature (Dunlap, 1999; Rosbash and Hall, 1989). Circadian clocks allow organisms to anticipate daily changes, temporally compartmentalizing diverse, and sometimes antagonizing, cellular processes (Hurley et al, 2014; Sancar et al, 2015b; Asher and Schibler, 2011; Hurley et al, 2018;

Baek et al, 2019), while the absence of a functional clock or the genetic or environmental perturbations of its function, can compromise organismal fitness and physiology (Roenneberg and Merrow, 2016).

Circadian clocks have appeared independently at least three different times throughout evolution and, consequently, all core-clock components do not share sequence conservation throughout the tree of life (Loudon, 2012; Rosbash, 2009; Dunlap and Loros, 2018). Yet, how these molecular gears interact in terms of circuitry is highly preserved across taxa (Rosbash, 2009; Dunlap, 1999). Indeed, circadian oscillators in fungi and animals are based on a transcription-translation negative feedback loop (TTFL) where Positive Elements (WC-1/WC-2 in Neurospora; CYC/CLK in Drosophila; BMAL1/CLOCK in mammals) directly drive the transcription of Negative Elements (FRQ in Neurospora; TIM/PER in Drosophila; CRY/PER in mammals) that nucleate multi-protein complexes—always including casein kinase 1 (CK1)—which feedback to inhibit the Positive Elements, thereby shutting down their own expression (Dunlap, 1999).

In Neurospora, White Collar-1 (WC-1) and White Collar-2 (WC-2) form the White Collar Complex (WCC) (Crosthwaite et al, 1997; Dunlap, 2006; Ballario and Macino, 1997), which in constant darkness binds the *clock box* (*c-box*) in the *frq* promoter (Froehlich et al, 2003), activating its expression. FRQ is produced, dimerizes, and interacts with the RNA helicase FRH (Cheng et al, 2005) and CK1, promoting the inactivation of WCC through its phosphorylation (Cheng et al, 2005; Schafmeier et al, 2005; He et al, 2006; Wang et al, 2019). FRQ is the substrate of several kinases, until it reaches a hyperphosphorylated state where it can no longer inhibit the WCC, allowing a new cycle of transcription to start; subsequently, hyperphosphorylated FRQ is degraded via the proteasome (He et al, 2003; He and Liu, 2005). Indeed, progressive phosphorylation of FRQ leading to its hyperphosphorylation and inactivation, and not degradation per se, appears to be critical for circadian cycling (Larrondo et al, 2015; Liu et al, 2019). In addition, WC-1 is a photoreceptor that in response to light induces the expression of many genes, including *frq* (Chen et al, 2009). In the presence of light, WCC recognizes a distinct *cis*-element within the *frq* promoter (*proximal light response element* or pLRE), boosting *frq* transcription, and allowing the synchronization of the clock

[1]Millennium Institute for Integrative Biology (iBio), Santiago, Chile. [2]Departamento de Genética Molecular y Microbiología, Facultad de Ciencias Biológicas, Pontificia Universidad Católica de Chile, Santiago, Chile. [3]Department of Mathematical Sciences, University of Cincinnati, Cincinnati, OH, USA. [4]Department of Pharmacology and Systems Physiology, University of Cincinnati, Cincinnati, OH, USA. [5]Department of Molecular and Systems Biology, Geisel School of Medicine at Dartmouth, Hanover, NH 03755, USA. [6]Department of Biochemistry and Cell Biology, Geisel School of Medicine at Dartmouth, Hanover, NH 03755, USA. ✉E-mail: lflarron@uc.cl

with the environment, in a process that is crucial for phase determination (Froehlich et al, 2002). Thus, the direct binding of the WCC to defined *cis*-elements within the promoter of the negative element *frq*, and the tight control over its regulation, is essential for proper circadian and light-induced expression of this core-clock component (Sancar et al, 2012; Belden et al, 2011, 2007b; Oehler et al, 2023).

In *Neurospora*, the central oscillator confers rhythmicity to diverse biological processes such as metabolism and conidiation (Dunlap, 1999). The information passes from the central oscillator to the output pathways, in part by a hierarchical arrangement of transcription factors that allows the rhythmic expression of an abundant cohort of clock-controlled genes (*ccgs*) (Hurley et al, 2016). *con-10* is one of those *ccgs*, with an undefined function in development and conidiation (Berlin and Yanofsky, 1985; Roberts et al, 1988; Ebbole et al, 1991), which also exhibits a strong and acute transcriptional response to light (Lauter and Yanofsky, 1993). Nevertheless, despite its strong photo-response, neither its induction by light nor its rhythmic expression have been reported to be directly controlled by the WCC (Hurley et al, 2014; Smith et al, 2010), and instead it has been described to depend on a complex regulation involving other transcription factors (Sancar et al, 2015a, 2015b, 2011).

One of the fascinating aspects about fungal and animal circadian clocks is how, despite their divergence from a phylogenetic origin a billion years ago, they display an evolutionarily conserved design: a one-step TTFL where Positive Elements directly control the expression of Negative ones. Moreover, as circadian oscillators started to be described across phyla, it was suggested that such one-step TTFL circuitry might be *"the only way you can make a clock"* (Barinaga, 1998). Provoked by such concept, we sought to challenge the genetic topological plasticity of a circadian TTFL and attest if an alternative circuit design could actually yield a functional oscillator. Thus, through transcriptional rewiring, we extended the original TTFL topology by subjecting *frq* expression to the control of a *ccg* promoter. We called this semi-synthetic design a hybrid oscillator (HO), as it combines core-clock components with cogs and gears that are normally part of the output (*ccg*) pathways. Due to the abundance of molecular tools, straightforward genetics, and the absence of gene families and paralogues, *Neurospora* is a great platform in which to adopt synthetic biology strategies, such as the implementation of new circadian clock topologies or synthetic circuits (Tabilo-Agurto et al, 2023; Matsu-Ura et al, 2018). Herein we characterize the HO generated by rewiring the *con-10* promoter to control *frq* expression, denominating it HO-10. HO-10 has a period close to 24 h in constant conditions, which can be dramatically shortened or lengthened by altering FRQ phosphorylation dynamics and, most remarkably, it is temperature compensated. Interestingly, other aspects such as responses to light and phase definition display novel and unexpected properties, revealing a "lights-on timer" behavior. Thus, this work uncovers that the evolutionarily conserved simple one-step TTFL is not the only possible functional circadian core-clock topology that could arise based on already existing cellular components. Analysis of this semi-synthetic oscillator also helps to underscore a key role for posttranslational regulation in properties such as temperature compensation and period determination, whereas transcriptional mechanisms appears as critical for clock phase determination and

entrainment to light, revealing also an unanticipated property such as the emergence of a lights-on timer logic.

## Results

### Generation of a functional hybrid oscillator

To change the architecture of the central oscillator, we eliminated the native connection between the Positive (WCC) and the Negative Element (*frq*) by a transcriptional rewiring strategy. Thus, using homologous recombination we replaced the endogenous *frq* promoter, including its *c-box*, *pLRE*, and 1.5 kbp 5' UTR, with different *ccg* promoters (including their respective 5' UTRs). As indicated above, we denominated these semi-synthetic circuits as hybrid oscillators (HOs), as they conserve parts of the wild-type oscillator and incorporate new components that innately only serve a role in output pathways (Fig. 1A). As proof of concept, we first chose three different *ccg* promoters already known to drive rhythmic transcription under constant conditions (constant darkness, DD), as confirmed by luciferase transcriptional reporters (Appendix Fig S1). Importantly, most *ccgs* are not only subjected to clock-regulation but also to a myriad of additional transcriptional inputs related to their specific role in the biology of the organism.

The capacity of the different HOs to generate and sustain rhythms in DD was evaluated using $frq_{c\text{-}box}$-*luc* which serves as a reporter for the activity of the TTFL Positive Element (WCC). Strains were grown in constant light (LL) for 24 h before monitoring them for luciferase activity in continuous darkness (DD). While WT strains showed strong and robust rhythms, the different rewired strains failed to exhibit a rhythmic behavior (Appendix Fig. S2). We then applied a LD12:12 entrainment protocol, for 3 days prior to their analyses in DD, observing that this allowed—particularly—one of the HOs to generate and sustain strong and stable oscillations under DD (Figs. 1B,C and EV1). We continued to characterize this HO, henceforth referred to as HO-10, as it was generated utilizing the promoter from the *ccg* known as *con-10*.

The expression of FRQ was confirmed in the different HOs through Western blot assays in LL and DD conditions (Appendix Fig. S3), observing that in the latter FRQ levels are three times higher in HO-10 than in the WT control, whereas in two of the arrhythmic HOs (generated with the promoters of *csp-1* and *medA*) FRQ levels were 5.2 and 6.5 times higher compared to WT, respectively (Appendix Fig S3). These results are compatible with the idea that in DD, high levels of FRQ such as the ones observed in the HOs with the *csp-1* or *medA* promoters, exert a constant inhibition of WCC as evidenced by low and arrhythmic levels of $frq_{c\text{-}box}$-*luc* expression (Appendix Fig. S2; Fig. EV1). In addition, as a negative control we also evaluated the effect of eliminating the native *frq* promoter, replacing it only by the selection marker cassette, confirming that when *frq* is not transcribed the system is arrhythmic, as seen in a *Δfrq*, which lacks its ORF (Appendix Figs. S2 and S3; Fig. EV1). We also examined FRQ levels in two additional HOs that exhibit oscillations, HO-*tub* and HO-*vvd*, generated with the *tubulin* and *vivid* promoter, confirming that FRQ levels in DD were similar (~1.1X) or lower (~0.5X) than the ones in HO-10 (Fig. EV2). HO-*tub* and HO-*vvd* were developed

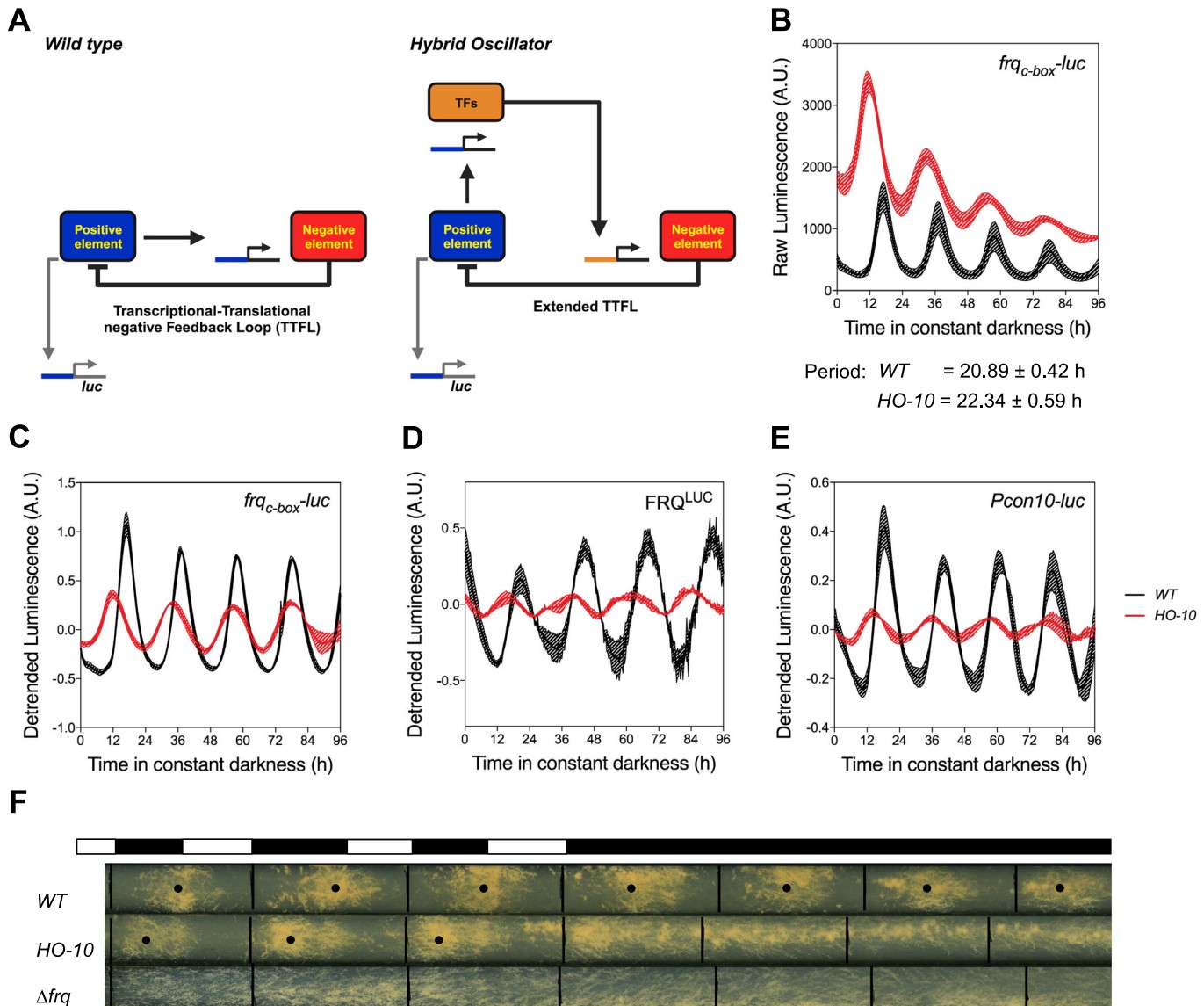

**Figure 1. A semi-synthetic oscillator sustains molecular oscillations, but not rhythmic overt conidiation under constant darkness.**

(A) Scheme of the native transcriptional translational feedback loop (TTFL) where the Positive Element (WCC, blue) directly regulates the transcription of the Negative Element (*frq*, red). To generate a semi-synthetic oscillator the promoter of the Negative Element was replaced by the promoter of a *clock-controlled gene* (*ccg*, orange), giving rise to a hybrid oscillator (HO) that combines core (blue and red) and output (orange) components in an extended TTFL topology. In both cases, a minimal *frq* promoter controlling *luc* expression serves as a readout of the system. (B–E) Evaluation of HO-10, constructed with the promoter of the *ccg* known as *con-10*, under constant darkness (DD) by analyzing LUC activity. Prior to luciferase monitoring in DD the strains were grown for 3 days under cycles of 12 h of light and 12 h of darkness (LD 12:12). The analyzed luciferase reporters were *frq_{c-box}-luc* (B) and (C), FRQ^{LUC} (D) and *P_{con10}-luc* (E). While in (B) raw data are shown, (C), (D), and (E) correspond to normalized detrended data. Each luciferase trace corresponds to the average of three different wells ± SD. All experiments were run three independent times, and a representative set is shown. (F) Race tube analysis of WT, HO-10, and a *frq*-less strain. Strains were analyzed for three days under LD 12:12 cycles and then transferred, on day four, to constant darkness. All experiments were run three independent times, and a representative set is shown. Source data are available online for this figure.

based on the promoters of a weakly and strongly rhythmic gene, respectively (Fig. EV2A,B). After LD entrainment HO-*vvd* exhibits only two peaks and then loses rhythmicity, whereas HO-*tub* shows low amplitude oscillations with an unstable period of 24.72 ± 3.06 h (Fig. EV2C,D).

In contrast, HO-10 exhibits robust oscillations with a period of 22.34 ± 0.59 h, which are slightly longer compared to the WT clock under these conditions (20.89 ± 0.42 h). In addition, we were able to confirm oscillations in FRQ protein levels utilizing a *frq^{LUC}* reporter

(Fig. 1D), providing additional evidence that the *con-10* promoter can yield *frq*/FRQ rhythmic expression. When utilizing a *P_{con10}-luc* reporter (Fig. 1E) we also observed that in HO-10 rhythms are being effectively passed to the output pathways, although overt rhythms in conidiation (a hallmark of *Neurospora* clock output) are not seen in DD when examined in race tube assays. The latter further confirms that—as inferred from different lines of evidence (Larrondo et al, 2015; Shi et al, 2007)—rhythms in *frq*/FRQ expression may not always be visualized by overt conidiation

rhythms. In addition, the lack of rhythmic conidiation in DD may be attributed to the decreased amplitude of the HO-10 oscillations. Nevertheless, HO-10 exhibits a cyclic conidiation pattern under a LD12:12 entrainment, similar to WT, contrasting the absence of periodic bands in a $\Delta frq$ strain (Fig. 1F).

As a proof of concept that HO-10 differs in its molecular circuitry from a WT clock, we assessed the consequences of eliminating an output regulator. We chose CSP-1, a TF known to be under clock control and that is part of the output pathways, but that it is not essential for clock function. *csp-1* encodes for a regulator that holds similarities to the yeast transcription repressors NRG1 and NRG2, and it has been described to have a large role in regulating metabolic genes in Neurospora (Sancar et al, 2011). Relevantly, it represents one of the best studied TF involved in circadian output, modulating a large number of *ccg*s, including *con-10*. In a WT clock devoid of *csp-1*, rhythms remain robust (Fig. EV3A) and period is unaltered under low sugar conditions (Sancar et al, 2012). In contrast, deletion of *csp-1* in HO-10 causes arrhythmicity (Fig. EV3B) confirming that—by definition—this output transcription factor is now a bona fide core-clock component and, therefore, part of the extended TTFL topology of HO-10.

### Period mainly depends on FRQ determinants

In order to further evaluate the contributions of different processes to period determination, we tested whether mutations known to affect FRQ phosphorylation-dynamics, and therefore period, would still do so in a context where *frq* transcriptional control had been dramatically changed. Thus, we reconstructed the HO-10 utilizing different *frq* alleles, some known to exhibit shorter ($frq^{\Delta C-term}$ and $frq^{S900A}$) or longer ($frq^7$ and $frq^{S538A, S540A}$) periods. As expected, such alleles behaved as reported in a WT oscillator background (Fig. 2A–G). Brilliantly, the same behavior was also evident when those alleles were tested in HO-10 (Fig. 2A–G). Unexpectedly, $frq^{5S\to D}$, which has been associated with the rapid degradation of this FRQ allele, fails to yield rhythms in a WT background but, nevertheless, behaved rhythmically in the context of the HO-10 (Fig. 2A–G; Appendix Fig S4). A similar behavior had been previously observed for $frq^{5S\to D}$ in a $\Delta fwd-1$ background (which exhibits impaired FRQ degradation) leading to overall high levels of this Negative Element (Larrondo et al, 2015). The latter observation is consistent with the fact that in HO-10 FRQ levels are higher, allowing the maintenance of rhythms of a highly unstable FRQ mutant such as $frq^{5S\to D}$.

Interestingly, independent of the *frq* allele that was analyzed, the first peak of the HO-10 always showed a phase advance and slightly longer period compared to its counterpart (Fig. 2) (see below). Thus, in the aggregate, the results indicate that while tampering with *frq*'s transcriptional control only appears to have a marginal effect on period, FRQ sequence changes known to modulate its phosphorylation dynamics and properties are, in return, a major variable impacting period length.

### Temperature compensation is unlikely a network-wide process

As temperature compensation is a defining property of clocks, we tested whether this was the case in HO-10, determining period at

22 °C, 25 °C, and 28 °C and comparing the $Q_{10}$ values for WT and HO-10. In our experiments we obtained a $Q_{10} = 0.96$ for the WT and a $Q_{10} = 0.94$ for the HO-10 (Fig. 3A), concluding that the rewired oscillator is also temperature compensated.

These results reinforce the hypothesis that temperature compensation is molecularly encoded in FRQ posttranslational events, such as phosphorylation (Mehra et al, 2009; Hu et al, 2021), and does not depend on particular aspects of *frq* transcription, or even on the characteristics of the *frq* 5'UTR. Indeed, it has been described that the latter undergoes temperature-dependent splicing that regulates the amounts of short (s-FRQ) and long FRQ (l-FRQ) isoforms, where the former leads to a longer period and l-FRQ to a shorter one (Diernfellner et al, 2007). In HO-10, *frq* 5'UTR is absent, so its splicing is unaffected by temperature and only l-FRQ is produced, as confirmed by western blot (Fig. 3B). Thus, HO-10 is temperature compensated even though it has an imbalance of short/long FRQ ratio, while also exhibiting an overall longer period. Interestingly, expression of only l-FRQ should render a shorter period, therefore the rather longer period in HO-10 is likely due to the introduction of additional steps/delays in the system. Besides its role in temperature finetuning of FRQ isoforms, the *frq* 5'UTR has several micro-ORFs that have been postulated to regulate mRNA levels and FRQ translation efficiency (Liu et al, 1997; Diernfellner et al, 2005; Colot et al, 2005). While, the absence of the entire *frq* 5'UTR in this rewired semi-synthetic oscillator argues against the assigned importance of this element in the clockworks, higher levels of *frq* transcription in HO-10 could partially compensate for such effects. In toto, and considering that the HO-10 possesses a very different (and intricate) circuitry topology, the results strongly indicate that transcriptional-wide processes are not playing a key role, as predicted by a network model of temperature compensation (Kurosawa and Iwasa, 2005), and that instead such clock property is expected to depend mainly on translational/posttranslational mechanisms.

### Unexpected properties of the semi-synthetic oscillator

Light is a main circadian input and, in *Neurospora*, it is signaled via the photoactivation of the WCC and its binding to the *pLRE* in the *frq* promoter, which leads to defined transcriptional dynamics (Froehlich et al, 2002; Oehler et al, 2023). Importantly, this region is no longer controlling *frq* expression in HO-10 and, moreover, a classic activation via WCC (at least with strong promoter binding) is not occurring in the *con-10* promoter (Smith et al, 2010; Hurley et al, 2014; Sancar et al, 2015a); therefore, in HO-10 light information is conveyed to *frq* by additional steps, involving other regulators downstream from WCC activity. Importantly, it had not escaped our attention that HO-10 reporter activity in DD, or race tubes under 12:12 LD cycles, revealed phase advances compared to WT. To further dig into this, we grew Neurospora on race tubes under different photoperiods (LD16:8, LD12:12, LD8:16, LD4:20, and LD0.5:23.5). Using the moment when lights are turned off as a reference mark, the phase of conidiation in WT is at the center of the photoperiod independent of the entrainment (Fig. 4A), whereas in HO-10 the conidiation band occurs at different times, depending on the photoperiod being tested (Fig. 4B). To analyze the effect of these different entrainment photoperiods in the phase of LUC levels in DD, we entrained strains for three days in such distinct photoperiods, and then we released them into DD. Monitoring

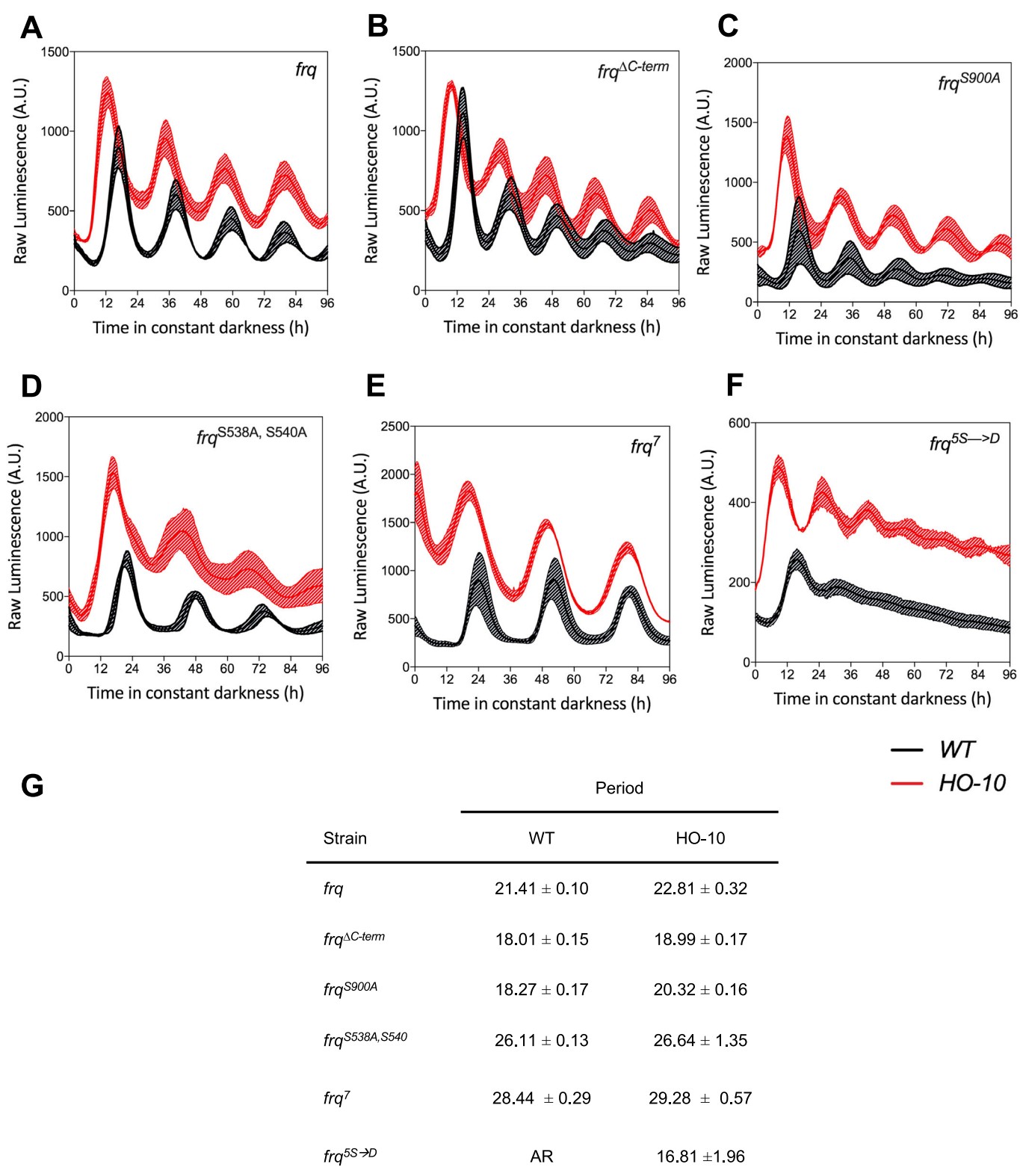

| Strain | Period | |
|---|---|---|
| | WT | HO-10 |
| *frq* | 21.41 ± 0.10 | 22.81 ± 0.32 |
| *frq*$^{\Delta C\text{-}term}$ | 18.01 ± 0.15 | 18.99 ± 0.17 |
| *frq*$^{S900A}$ | 18.27 ± 0.17 | 20.32 ± 0.16 |
| *frq*$^{S538A,S540}$ | 26.11 ± 0.13 | 26.64 ± 1.35 |
| *frq*$^{7}$ | 28.44 ± 0.29 | 29.28 ± 0.57 |
| *frq*$^{5S\rightarrow D}$ | AR | 16.81 ± 1.96 |

AR: Arrhythmic

**Figure 2. The period of the semi-synthetic Oscillator HO-10 is dependent on FRQ determinants.**

(A–F) Different *frq* alleles, as indicated in the insets, were analyzed in the context of a WT (black) or the HO-10 semi-synthetic circuitry (red), under constant darkness (DD), utilizing *frq_{c-box}-luc* as a reporter. (A–F) Samples were entrained for three days under 12:12 LD cycles, prior to monitoring. In all cases, experiments were run three independent times, and a representative set is shown. Each luciferase trace corresponds to the average of three different wells ± SD. (G) Period lengths of the different *frq* alleles were calculated for WT and HO-10 strains. Source data are available online for this figure.

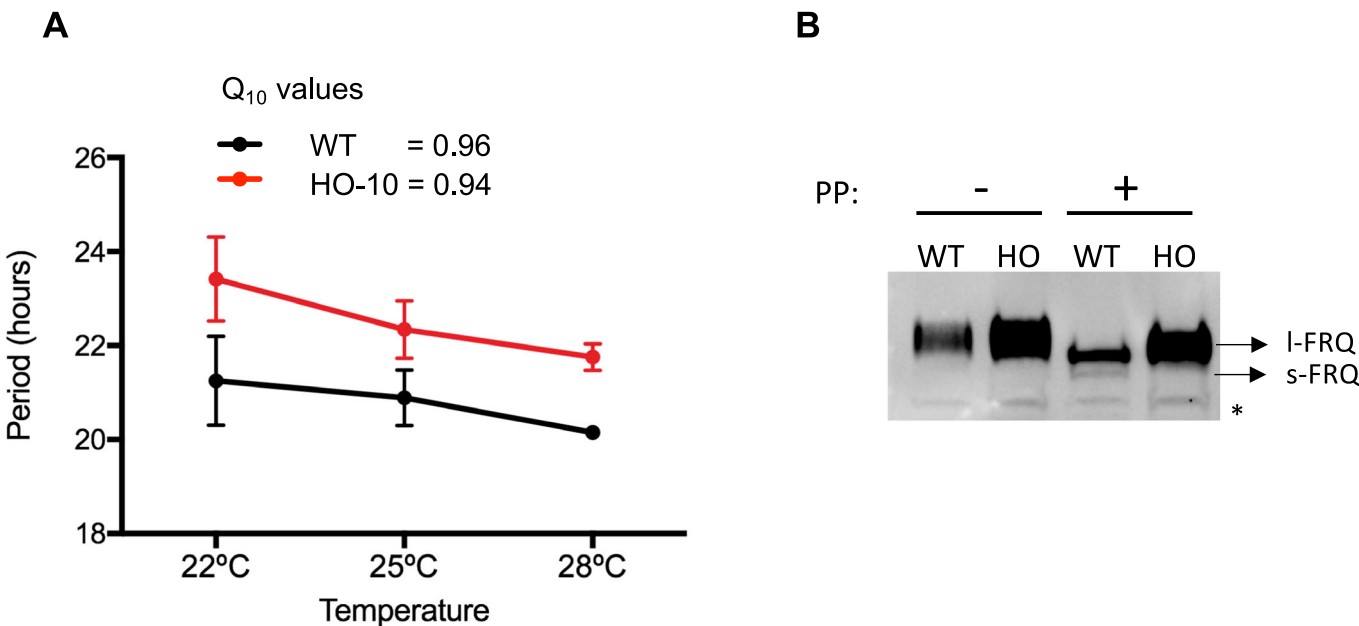

**Figure 3. The semi-synthetic oscillator HO-10 is temperature compensated.**

(A) Strains were grown for three days under LD 12:12 cycles at 25 °C, and then monitored in DD at three different temperatures 22 °C, 25 °C, and 28 °C, utilizing *frq_{c-box}-luc* as a reporter, and period and Q values were calculated. Data leading to $Q_{10}$ calculation were plotted as the average ± SD of three different measurements. (B) To evaluate the presence of s-FRQ and l-FRQ, strains were grown for 48 h in LL at 25 °C and proteins were extracted and treated with phosphatase, prior to SDS PAGE in the WT and HO-10 (HO), observing that the latter only expresses l-FRQ. * Unspecific band. Source data are available online for this figure.

LUC activity, we observed that while WT exhibits the same phase after all entrainments (Fig. 4C), in HO-10 the phase varies according to the preceding photoperiod (Fig. 4D). These results demonstrate that like WT, the HO-10 is (i) capable of perceiving environmental information and (ii) retaining it after transfer to constant conditions, both being defining critical properties of circadian clocks. But remarkably, HO-10 shows a distinct difference compared to the WT oscillator. The latter fixes its phase at the moment lights are turned off, which can be clearly seen as a slope close to 0 when the phase of conidiation or LUC activity is plotted relative to photoperiod, indicating dusk dominance (Edwards et al, 2010). In contrast, HO-10 displays a slope close to −1 denoting a dawn dominance, which implies that it fixes its phase when the lights are turned on (Fig. 4E,F). To further confirm the concept that phase in HO-10 is determined at the moment lights are turned on, new experiments were conducted such that luciferase tracking was synchronized to the moment when lights would be turned on after the different entrainment regimes. When doing that and, as expected, LUC expression in HO-10 shows the same phase, whereas the phase of the WT clock is instead broadly distributed (Appendix Fig S5). Finally, we also performed the same experiments but now using race tubes, and providing reference marks at the lights-on transitions, observing that relative to these marks the conidial

bands exhibit the same phase in HO-10, but not in WT (Appendix Fig. S5). In summary, these results indicate that the semi-synthetic HO-10 clock works with the logic of a lights-on timer.

Since HO-10 revealed particular ways of processing light information, we also evaluated the effect of a discrete saturating 30 min light pulse (LP) (Crosthwaite et al, 1995). Importantly, the LP was applied on strains that have been in DD for 48 h, but which displayed different circadian times (CT). The LP produces clear phase shifts in the WT oscillator (Fig. 4G), whereas the clock in HO-10 is completely reset to early subjective day independent of the CT at which the pulse was given (Fig. 4H). Thus, while a WT clock exhibits a type 1 Phase Transition Curve (PTC), the hybrid HO-10 displays a type 0 PTC (Winfree, 1970). This was true for short LP of even 5 min under our tested conditions, yet when a 1 min LP was administrated the HO-10 exhibited a type 1 PTC (Fig. EV4). This may be explained by the fact that in response to light the *con-10* promoter has a stronger induction pattern than *frq*'s native one (Tan et al, 2004; Wu et al, 2014). Notably, the effect of a LP was also evaluated when HO-10 was in an arrhythmic state, such as when transferred to DD after only 24 h in LL (Appendix Fig. S2) evidencing that a single 30 min LP is capable of taking this semi-synthetic oscillator from an arrhythmic to a rhythmic orbit (Fig. EV5A). Likewise, LPs of 5 min or longer are capable of

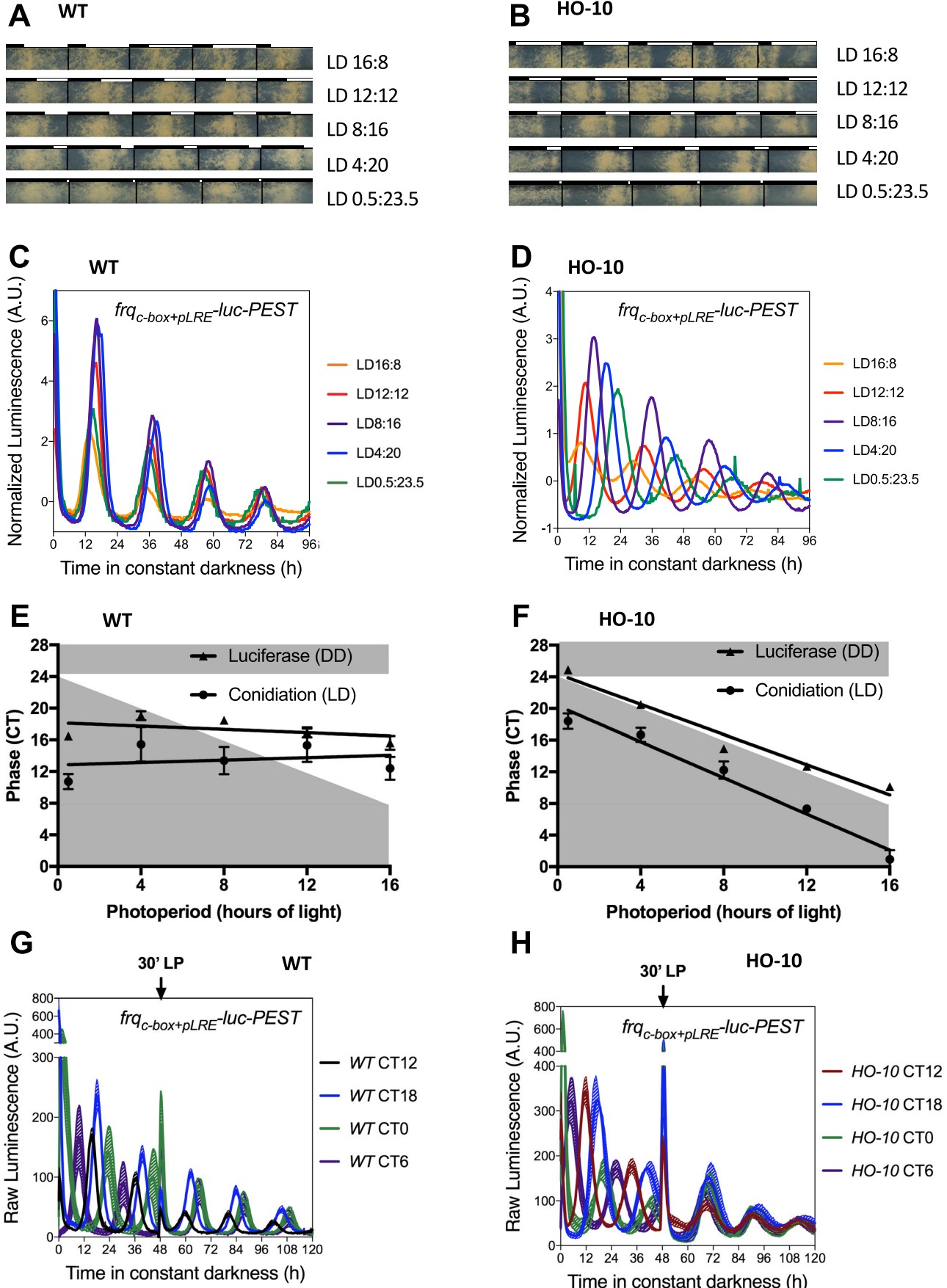

◄ **Figure 4. The semi-synthetic HO-10 clock presents unexpected responses to light stimuli.**

(A,B) The phase of HO-10 dramatically changes depending on the LD regime, as observed by analyzing race tubes grown under the indicated LD conditions, where daily marks were placed at the light to dark transitions. Experiments were run three independent times, and a representative set is shown. (C,D) Luciferase activity was monitored as cultures were transferred to DD after 3 days of the indicated entrainments. Experiments were run three independent times, and a representative set is shown. Each luciferase trace corresponds to the average of three different wells. (E,F) Plots depicting the phase (with symbols denoting average, and error bars the SD) of conidiation (race tubes) under LD conditions (A) and (B), and of luciferase activity under DD conditions after the different entrainments (C) and (D). (G,H) WT and HO-10 strains were entrained for three days under LD 12:12 cycles in four different incubators to set four different phases prior transferring to DD, when monitoring of luciferase started. After 48 h in DD a 30 min LP was applied, such that strains were at different circadian times (CTs) when the pulse was given. $frq_{c\text{-}box+pLRE}$-luc-PEST was used as reporter. Experiments were run three independent times, and a representative set is shown. Each luciferase trace corresponds to the average of three different wells. Source data are available online for this figure.

generating strong oscillations, when the system is in an arrhythmic state (Fig. EV5). All this indicates that the HO-10 clock has a hypersensitivity to light stimuli.

## Mathematical modeling of the semi-synthetic HO-10

Finally, we modified the mathematical model implemented by Dovzhenok and cols (Dovzhenok et al, 2015), to better understand the behavior of this semi-synthetic oscillator under different entrainment conditions. Briefly, we abstracted the wiring of HO-10 as described in Fig. 1A, and considered different light-dependent kinetics of *con-10* vs *frq* promoters based on reported time-series data (Tan et al, 2004). Specifically, *con-10* mRNA shows rapid induction followed by photoadaptation when *Neurospora crassa* is transferred from dark to light. Hence, we assumed that the rate of *con-10* expression ($k_{24}$) undergoes transient light-dependent increase. Importantly, such abstraction allowed us also to prescind of the exact topology of the HO-10 circuitry. Our analysis indicates that HO-10 resides in a non-oscillatory domain when it's transferred from LL to DD, since the rate of *frq* transcription is significantly reduced in LL due to the lower activity of *con-10* promoter, which is caused by its rapid photoadaptation (Lauter and Yanofsky, 1993; Tan et al, 2004). However, a 30 min LP in DD boosts the activity of this promoter increasing the rate of *frq* expression (Fig. EV5), which can push the system into an oscillatory domain (Fig. 5A,B). Specifically, a light pulse increases the rate of *con-10* expression ($k_{24}$) pushing the system from a stable steady state to an unstable steady state with a stable periodic limit cycle domain enabling autonomous oscillations (red arrow, in Fig. 5B). On the other hand, if HO-10 is grown in LL, then the rate of *con-10* expression ($k_{24}$) is decreased due to the lower activity of *con-10* promoter moving the system to a region of stable steady states (blue arrow, Fig. 5B). Based on this model, we predicted that a HO-10 strain deficient in photoadaptation, such as *Δvvd*, would be rhythmic after a direct LL to DD transfer, as *frq* levels would be higher in LL, allowing efficient inhibition of WCC once in DD, a prediction that was successfully confirmed (Fig. 5C).

Furthermore, mathematical modeling of this rewired oscillator, also provided unexpected insights into the understanding of the WT system, particularly regarding how transcriptional regulation of the core oscillator modulates light-dependent responses. CSP-1 is a known light-inducible transcriptional repressor, and our computer simulations revealed that light-induced CSP-1 exerts stronger repression on WC-1 in LD12:12 compared to LD4:20 (Appendix Fig. S6A,B). This results in reduced expression of *frq* mRNA and nuclear FRQ (FRQn) in LD12:12 compared to LD4:20, and a phase shift when comparing these two entrainment

conditions (Appendix Fig. S6A,B). In contrast, HO-10 shows no phase difference under the two entrainment regimens due to the strong light induction of the *con-10* promoter driving the expression of *frq* in the hybrid oscillator (Appendix Fig. S6C,D). The above data suggest a potential role of CSP-1 regulating the phase of circadian rhythms. Hence, we tested in silico the consequences of not having *csp-1* with respect to phase information. Our simulations indicate that removal of *csp-1* in the model results in a phase shift that resembles the behavior of HO-10 in DD conditions. Specifically, in *Δcsp-1* we observe a phase difference of ~8-h under free-running DD conditions after LD12:12 and LD4:20 entrainment regimens (Fig. 5D; Appendix Fig. S7). To validate the above model prediction, we tested conidiation rhythms using race tubes of *Δcsp-1* under free-running conditions after LD12:12 and LD4:20 entrainment regimens. Our experimental data confirmed that, as predicted in our model, a LD12:12 regime causes a significant phase advance compared to LD4:20 (~3 h), albeit of a smaller magnitude than expected (Fig. 5E; Appendix Fig. S7). These data suggest that the underlying transcriptional regulation of the WT system includes intricate aspects controlling light responses via the *frq* promoter, integrating signals from WCC and CSP-1. While CSP-1 had already been implicated in metabolic compensation, these new results uncover its unexpected role in modulating *frq* expression and subsequent phase changes of circadian rhythms in different photoperiods. Such role could be the sum of CSP-1 effects on the *wc-1* and *frq* promoters, where the absence of *csp-1* is evidenced by an altered phase in DD following distinct entrainment conditions. Notably, the behavior of a WT clock devoid of *csp-1* resembles HO-10, which due to its rewired transcriptional circuitry processes light information in a different fashion having, as a consequence, altered phase responses. Importantly, this ancillary role of CSP-1 in the WT core-clock contrasts the essential role that CSP-1 plays in the HO-10 oscillator.

# Discussion

While several efforts have sought to establish fully functional synthetic circadian oscillators, this has proven a daunting task in part due to the underlying complexity of the clockworks (Elowitz and Leibler, 2000; Purcell et al, 2010). Instead of building a fully synthetic clock, our transcriptional rewiring approach allowed implementing a semi-synthetic one, which we denominated hybrid as it combines, as core-components, parts from the native oscillator and elements from the output pathways. Thus, the results show that we can significantly modify the circuit topology of an evolutionarily conserved circadian TTFL and still have a functional clock. In this

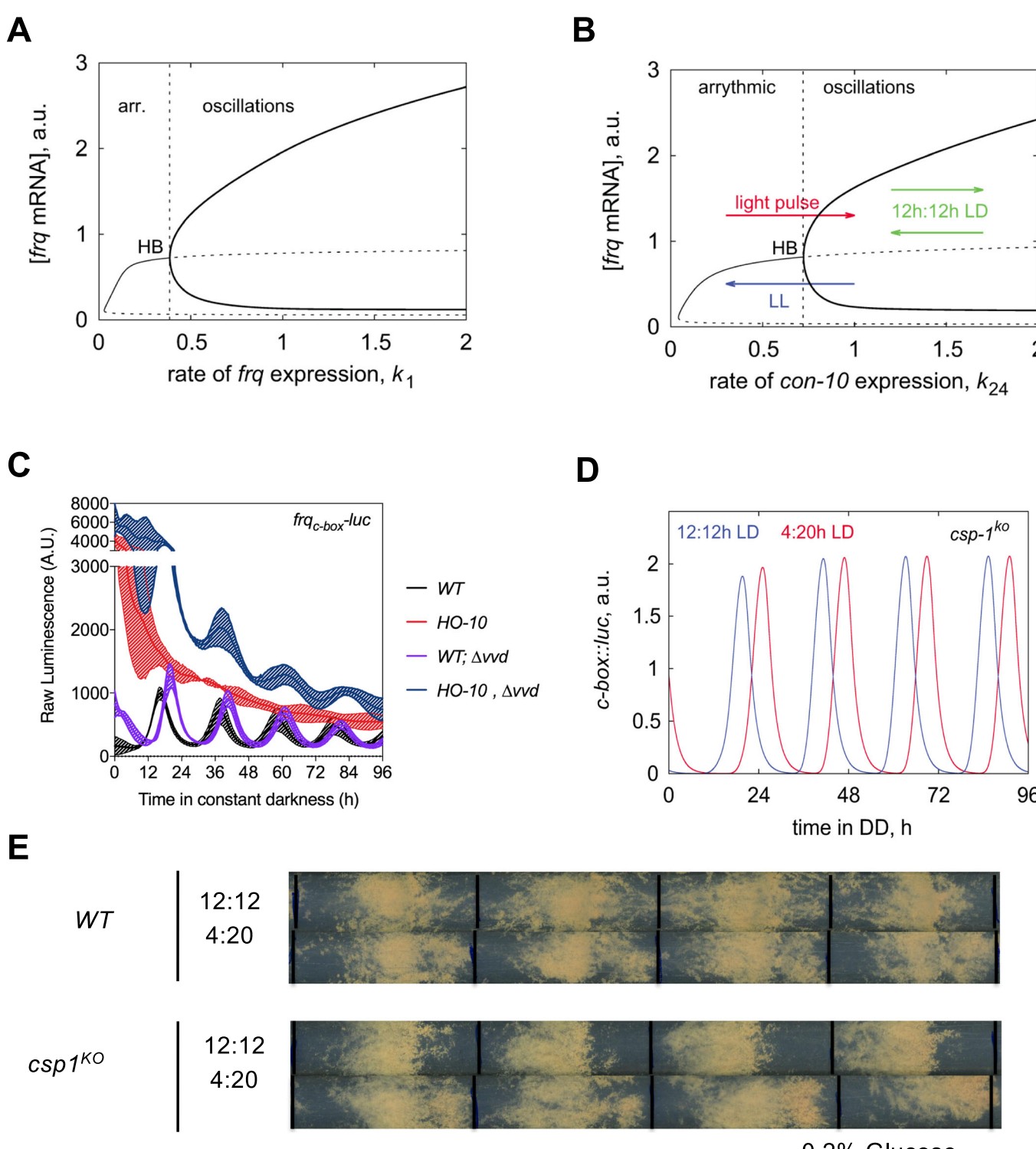

new architecture the canonical *c-box* and *pLRE* sequences (key in regulating native *frq*) are no longer controlling its expression and importantly, *frq* transcription (commanded now by the *con-10* promoter) ceases to be simply and directly regulated by the WCC. Although multiple ChIP-Seq datasets, including our own unpublished ones, have failed to detect WCC occupancy at the *con-10* promoter either under DD conditions or after a light-pulse (Smith

et al, 2010; Hurley et al, 2014; Sancar et al, 2015a), we cannot fully discard a transient action of WCC on the *con-10* promoter (i.e., WCC acting as a pioneer TF facilitating the recruitment of additional components). Thus, a limitation of the current study is that part of the resulting circuitry may still relay on a direct effect of WCC, which would mean that the core-clock topology would be an admixture of a classic TTFL circuit, entangled with multiple output

◄ **Figure 5. A mathematical model helps explaining the properties of the HO-10 and to predict new properties of the WT oscillator.**

(A,B) Bifurcation diagrams for expression rates, where vertical dashed lines represent the critical expression rates that separate arrhythmic from oscillatory regions (HB = Hopf bifurcation). Thin solid (dashed) curves represent stable (unstable) steady states of the model. Thick black curves show the envelope (max and min) of the oscillatory solution. Diagrams for *frq* expression rate in the WT clock (**A**) and HO-10 (**B**). The blue arrow represents the reduction of *con-10* expression below the critical value when HO-10 is transferred from LL to DD. The red arrow represents the strong activation of *con-10* expression when HO-10 is subjected to a LP, pushing HO-10 into the oscillatory region. The green arrow represents HO-10 in the oscillatory region when is entrained using 12:12 LD. (**C**) The strains were grown for 24 h in LL and transferred to DD where LUC activity coming from a *c-box-luc* reporter was evaluated. The black traces represent a WT strain, in purple Δ*vvd*, in red the HO-10 and in blue the HO-10, Δ*vvd*. Each luciferase trace corresponds to the average of three different wells ± SD. (**D**) Model simulations predict that the phase-correcting mechanism in the WT clock is CSP-1-dependent. (**E**) Experimental data (based on three independent experiments, of which a representative set is presented) confirms the model prediction (**D**). The strains were grown in race tube with 0.2% glucose medium under the indicated light regime LD 12:12 or LD 4:20. Every 24 h the growth front was marked. Source data are available online for this figure.

nodes. Nevertheless, it is clear that the rewired TTFL presents dynamics that are quite different from the canonical *frq* promoter (given by its native elements), which exhibits distinct refractoriness or complex transcriptional and chromatin remodeling dynamics (Oehler et al, 2023; Cesbron et al, 2015). Future efforts will be focused on providing a detailed transcriptional landscape of this rewired oscillator, including the main TFs acting as bona fide positive elements, particularly unravelling the exact direct contribution of WCC. As shown in Fig. EV3, deletion of a TF known to be part of the output pathways, such as CSP-1, confirms that although this TF is not essential for a WT clock, its absence causes arrhythmicity in the context of HO-10: in other words, CSP-1 is now, by definition, a genuine core-clock component. Although FRQ can still directly inhibit WCC activity, the latter now also controls transcription of *frq* via output components that act upon the *con-10* promoter, such as CSP-1, and likely many clock-controlled TFs such as SUB-1 (Sancar et al, 2015a). Notably, despite this altered topology, circadian behavior is still achieved. Importantly, as hinted earlier, this rewired architecture scrambles several essential parameters described as key to achieving proper clock function such as the transcriptional rates of the Negative Element (Froehlich et al, 2002), post-transcriptional regulation mediated by *frq* 5'UTR (Liu et al, 1997; Garceau et al, 1997), chromatin remodeling processes associated with the *c-box* (Wang et al, 2014; Gai et al, 2017; Belden et al, 2011) and, most remarkable, the basic and broadly conserved topology of eukaryotic circadian TTFLs (Ode and Ueda, 2018). It is notable that systematic deletion of different regulators in the WT oscillator, have failed to reveal any other TF (besides WC-1 and WC-2) essential for the clockworks (Muñoz-Guzmán et al, 2021), which contrasts with the acquired crucial role of CSP-1 in HO-10 rhythmicity. It is also noteworthy that one of the other semi-synthetic oscillators tested in this work (HO-*tub*) exhibited an oscillatory behavior (Fig EV2), despite bearing a promoter that is loosely connected to the output pathways, reinforcing the idea of the genetic plasticity of a functional circadian oscillator. In contrast HO-*vvd*, which is based on the *vvd* promoter (a direct target of WCC that displays rhythmic transcription (Cesbron et al, 2013)), exhibits limited oscillations, reflecting that it is not all just about the wiring of the circuit, but also dependent on the transcriptional landscape and kinetics governing expression of the Negative Element (Oehler et al, 2023). In our current limited experience, period of the HOs appears to be longer than WT, and although the two peaks seen in HO-*vvd* may be suggestive of a shorter period, we cannot discard the effect of transients that may obscure proper period determination. We

foresee that detailed study of these and other semi-synthetic circuits will continue to yield important circadian insights.

Indeed, the characterization of a functional HO, such as HO-10, allows separation or decoupling of clock properties that are determined by transcriptional- versus posttranslational- mechanisms. In this work we have shown that phase determination and sensitivity to light responses are highly dependent on transcriptional mechanisms, whereas period determination and temperature compensation are mainly dictated by posttranslational mechanisms. Indeed, it is extremely informative to confirm that a fascinating property such as temperature compensation remains intact in HO-10. These results further indicate that this emergent property is unlikely to depend on a transcriptional network property, as expected from a network model of temperature compensation (Kidd et al, 2015; Kurosawa and Iwasa, 2005) and that, instead, it strictly relies on posttranslational mechanisms (Hu et al, 2021; Mehra et al, 2009; Wang et al, 2023).

Another interesting observation is that HO-10 exhibits robust molecular oscillations, whereas overt conidiation rhythms are impaired, as seen also in some other mutants (Larrondo et al, 2015; Shi et al, 2007). Thus, the core-oscillator appears more resilient to changes in state variables compared to the clock output, which could be interpreted as that the oscillator cogwheels need to be of a minimal size (i.e., amplitude) to properly engage with the cogs controlling overt output. While we have interpreted our results based on relative FRQ levels, we are not oblivious to the fact that quality of FRQ (i.e., timing on phospho-forms) may be, potentially, playing even stronger effects.

Importantly, although the semi-synthetic oscillator is still able to perceive and entrain to light cues, it processes this information with different dynamics, behaving as a lights-on instead of a lights-off timer. Indeed, the HO demonstrates that to switch the same basic oscillator from dusk to dawn-synchronization it is necessary only to strongly repress or modify the direct action of light on the Negative Element (i.e., *frq* pLRE), and to replace this with a new controller that depicts very distinct light-dynamics (*con-10* promoter). The co-existence of dawn and dusk-synchronized clocks in different cells of the same organism has been noted both in plants (e.g., Edwards et al, 2010) and in the SCN in mammals (e.g., Inagaki et al, 2007), and it may be that the HO provides insight into how this could be achieved. The initial action of light to reset the mammalian clock is the rapid light-induction of *Per1* (Shigeyoshi et al, 1997). If this transduction pathway was repressed in a cell-type-specific manner and replaced by rapid light-induction of a transcription factor that secondarily activates *Per1* (analogous to

the replacement of the simple direct action of WCC on *frq* seen in WT *Neurospora* with a more complex action of light on *con-10* which also secondarily acts on *frq* in the HO), an oppositely-synchronized clock might emerge. With no doubt, the molecular dissection of semi- or/and fully-synthetic oscillators will help further uncovering the clockworks of timing-circuits. Such an approach should also provide a unique opportunity to examine the physiological impact of changing clock systemic properties, such as phase determination, eventually leading to the proximate and ultimate causes underlying phase selection in a given species.

Finally, HO-10 constitutes in itself evidence that other circuitries (different from a one-step closed TTFL, and based fully on rewiring endogenous components), can indeed act as functional circadian oscillators. Therefore, from a topological perspective, there are different ways of making a clock yet, by parsimony, evolution appears to have always chosen the simplest design.

# Methods

## Plasmids and strains

All the plasmids were constructed using yeast recombination cloning using PCR amplification products, and the *Neurospora* strains were transformed by electroporation with the dialyzed PCR products obtained with Phusion Flash, following a similar protocol as already described (Colot et al, 2006). The primers used to generate the transcriptional reporters are detailed in Table EV1. To generate the different rewired strains to be tested as HOs, we eliminated the control of *frq* by its native promoter, and replaced it by selected *ccg*s promoters. The primers used to create the constructs and plasmids are detailed in Table EV2. The allelic replacement of the *frq* promoter was conducted as described (Larrondo et al, 2009), and correct integration of the genetic constructs were confirmed by PCR.

*Neurospora* was grown at 25 °C in constant light (LL) on slants with minimal Vogel's 1X media supplemented with 2% sucrose w/v and 1.5% agar w/v (Vogel, 1956). The strains x654-14a (*ras-1^bd^; mus-51^rip^; his-3::frq_{c-box}-luc, a*), x658-8a (*ras-1^bd^; mus-51^rip^; frq^LUC^; a*), xc1783-4a (*ras-1^bd^; mus-51^rip^; csr-1::frq_{c-box+pLRE}-luc_PEST, a*) and x383-4A (*ras-1^bd^; mus-51^rip^; his-3::Pcon10-luc; A*) were utilized as the recipient strains for the rewiring of *frq* expression, whereas x654-1a (*ras-1^bd^; mus-51^rip^; a*) was used to generate the *promoter-luc* reporters.

The different strains containing *frq_{c-box}-luc* and different *frq* alleles (*frq^V5^; frq^ΔC-term^; frq^S900A^; frq^S538A,S540A^* and *frq^5S->D^*) were previously described (Larrondo et al, 2015), and herein were crossed to x654-16A (*ras-1^bd^; mus-51^rip^; his-3::frq_{c-box}-luc, a*), in order to obtain xc2116-10 (*frq^V5^, ras-1^bd^; mus-51^rip^; his-3::frq_{c-box}-luc, a*), xc2117-12 (*frq^S900A^, ras-1^bd^; mus-51^rip^; his-3::frq_{c-box}-luc, A*), xc2118-6 (*frq^S538A,S540A^, ras-1^bd^; mus-51^rip^; his-3::frq_{c-box}-luc, a*), xc2119-1(*frq^ΔC-term^, ras-1^bd^; mus-51^rip^; his-3::frq_{c-box}-luc, a*), xc2121-7(*frq^5S->D^, ras-1^bd^; mus-51^rip^; his-3::frq_{c-box}-luc, a*). The *frq^7^* strain was obtained by a sexual cross between x578-9 (*his-3::frq_{c-box}-luc, ras-1^bd^, frq^7^, a*) and x654-7A (*ras-1^bd^; mus-51^rip^; A*) to obtain strain xc1755-3a (*ras-1^bd^, mus-51^rip^, his-3::frq_{c-box}-luc, frq^7^, a*). These strains were utilized to build the HO-10 (replacing the *frq* promoter by the one of *con-10*), in genetic backgrounds bearing different *frq* alleles.

## Homokarionization by microconidiation

To avoid ripping (Selker and Garrett, 1988; Watters et al, 1999), the HOs were homokarionized by microconidiation. *Neurospora* was inoculated on a slant with 6 mL of microconidiation media (0.5% sucrose w/v, 0.1X Westergaard w/v, 2% agar w/v) supplemented with fresh sterile 60 μl of iodoacetate 0.1 M. Strains grew for 12 days in 12:12 LD cycle previous harvest using 2 mL of sterile water and filter with a 5-μm pore size syringe filter (EMD Millipore™ SLSV025LS). 150 μl were plated with the corresponding selective media, grew overnight at 30 °C and colonies were picked and transferred to slant with the selective media. Homokarionization was confirmed by PCR with the primers detailed in Table EV3.

## Culture conditions

In vivo bioluminescence was conducted, as already reported (Larrondo et al, 2015; Muñoz-Guzmán et al, 2021) in 96-wells plates with LNN-CCD media (0.03% glucose, 0.05% arginine, 50 ng/ml biotin, 1.5% agar, and 25 μM luciferin) supplemented with quinic acid (QA 0.01 M) at 25 °C, unless otherwise specified. The different LD entrainments are indicated in each figure. Strains for Western blot were grown in *petri* dishes with LNN-CCD + QA as in (Larrondo et al, 2015). Race tube analyses were conducted as reported (Belden et al, 2007a) and for *Δcsp-1* experiments media included glucose 0.2%. All experiments were performed in Percival incubators equipped with white cool light fluorescent tubes (light intensity up to 100 μM/m²/s; wavelength 400–720 nm).

## Luciferase-based analysis

Strains were incubated according to the entrainments specified in each experiment. Data acquisition was conducted as described (Larrondo et al, 2015). Period and phase analyses were performed in BioDare2 (Zielinski et al, 2014) as previously indicated (Muñoz-Guzmán et al, 2021). For strains of different genotypes, several clones (in general 3) were selected, after which a representative one was utilized for the different analyses. Experiments were performed at least three independent times and in each one samples were inoculated in triplicate. When plotted each line corresponds to the average of three different wells ± SD.

## DNA analysis

DNA extraction was performed as previously described (Cenis, 1992), but using conidia as starting material. All strains were confirmed by PCR.

## Protein extraction and western blotting

Proteins were extracted by TCA method and Western blot was performed loading 40 μg of total protein in 4–20% Mini-PROTEAN® TGX™ precast protein gels and FRQ antisera was used. Phosphatase treatment was performed as manufacture instructions (New England Biolabs p0753s). All experiments were performed three times.

## Mathematical modeling

The Neurospora circadian clock model (Dovzhenok et al, 2015) was adapted to include genes downstream of the core clock. An

unknown transcription factor X (*tfx*) mRNA expression was included with the rate constant $k_{20}$ ($k_{20} = 1.5\,\text{h}^{-1}$), TFX protein synthesis with the rate constant $k_{22}$ ($k_{22} = 5\,\text{h}^{-1}$), and *con-10* mRNA expression with rate constant $k_{24}$ ($k_{24} = 1\,\text{h}^{-1}$). *tfx* mRNA, TFX, and *con-10* mRNA degrade with the rate constants $k_{21}$, $k_{23}$, and $k_{25}$, respectively ($k_{21} = 2.8\,\text{h}^{-1}$, $k_{23} = 2.8\,\text{h}^{-1}$, $k_{25} = 2.8\,\text{h}^{-1}$).

To model the hybrid oscillator, the promoter of *frq* was substituted with the promoter of *con-10*. This modification rewires and extends the core negative feedback loop that drives circadian oscillations to include genes downstream of the core clock (such as *tfx*). All the mathematical equations and model parameters are detailed in Extended methods.

Mathematical modeling was carried out using XPP-AUT computer program (Ermentrout, 2002).

Code is available upon request.

## Data availability

All data are available in the main text and the expanded view materials. Raw data files are available upon request.

## Peer review information

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

## Acknowledgements

This work was founded by ANID-Millennium Science Initiative Program-Millennium Institute for Integrative Biology (iBio ICN17_022), grant number ANID/FONDECYT 1211715, the International Research Scholar Program of the Howard Hughes Medical Institute, and The Richard Lounsbery Foundation. Additional funding was provided by grant number ANID/FONDECYT Postdoctorado 3220747 to AG. JCD and JJL were supported by NIH grants R35GM118021 and R35GM118022, respectively.

## Author contributions

**Alejandra Goity**: Conceptualization; Funding acquisition; Investigation; Visualization; Methodology; Writing—original draft; Writing—review and editing. **Andrey Dovzhenok**: Resources; Investigation; Visualization; Methodology; Writing—review and editing. **Sookkyung Lim**: Investigation; Visualization; Methodology; Writing—review and editing. **Christian Hong**: Supervision; Funding acquisition; Investigation; Visualization; Methodology; Writing—review and editing. **Jennifer Loros**: Supervision; Funding acquisition; Writing—review and editing. **Jay C Dunlap**: Supervision; Funding acquisition; Writing—review and editing. **Luis F Larrondo**: Conceptualization; Supervision; Funding acquisition; Methodology; Writing—original draft; Project administration; Writing—review and editing.

## Disclosure and competing interests statement

The authors declare no competing interests.

# Expanded View Figures

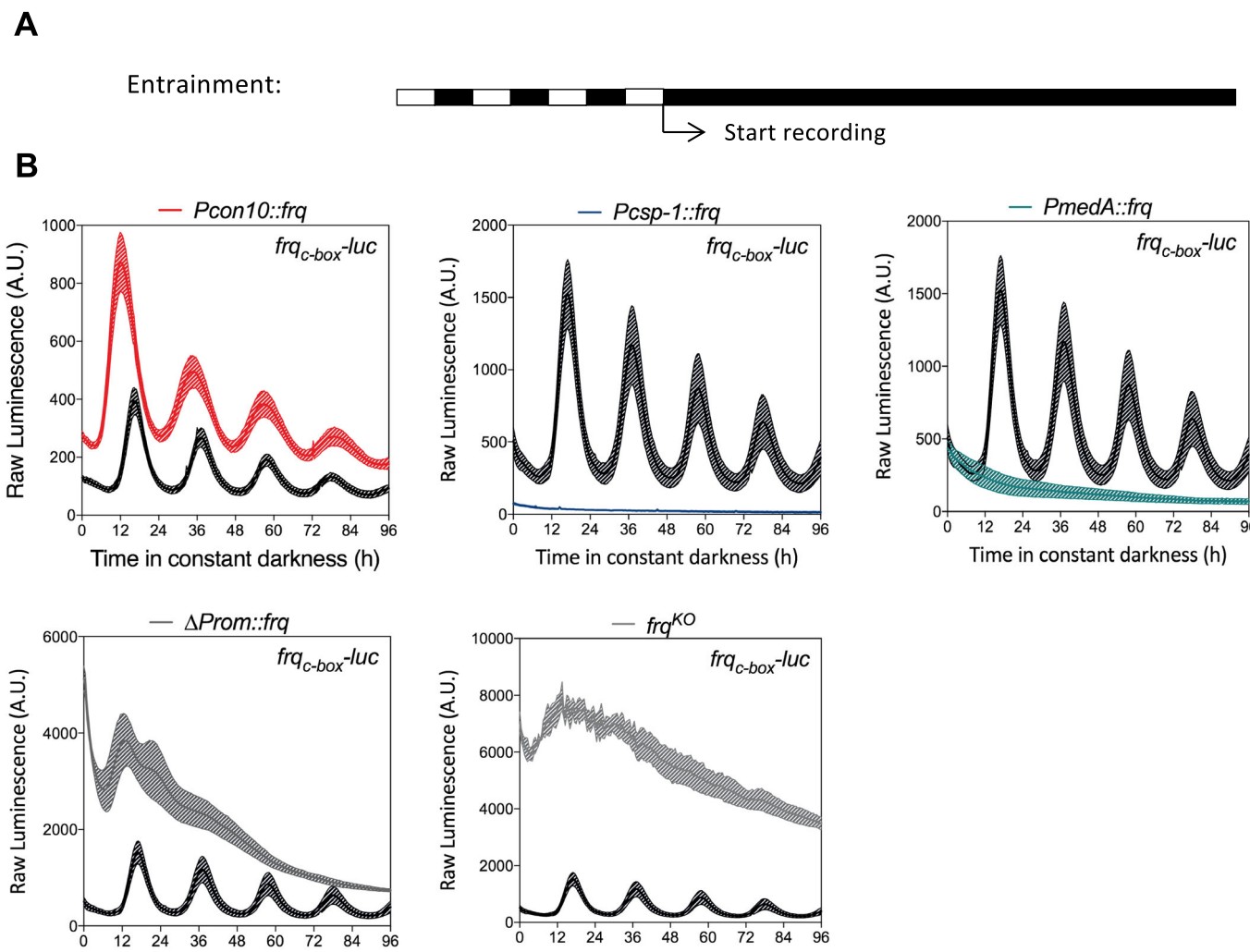

**Figure EV1. Only one the tested Hybrid Oscillators sustains robust rhythms after a three days LD 12:12 entrainment.**

(A) Entrainment protocol used to evaluate the different HOs. Prior to recording in DD, the strains were entrained for three days under 12:12 LD cycles. (B) Evaluation of HOs under DD conditions, by analyzing LUC activity coming from a $frq_{c-box}$-*luc* reporter. The black traces represent a *wild type* strain, whereas the different HOs are depicted in color. A negative control without a promoter (Δ*Prom::frq* only containing the resistance cassette, *bar*), as well as a $frq^{KO}$ were examined. In all cases, experiments were run three independent times, and a representative set is shown. Each luciferase trace corresponds to the average of three different wells ± SD.

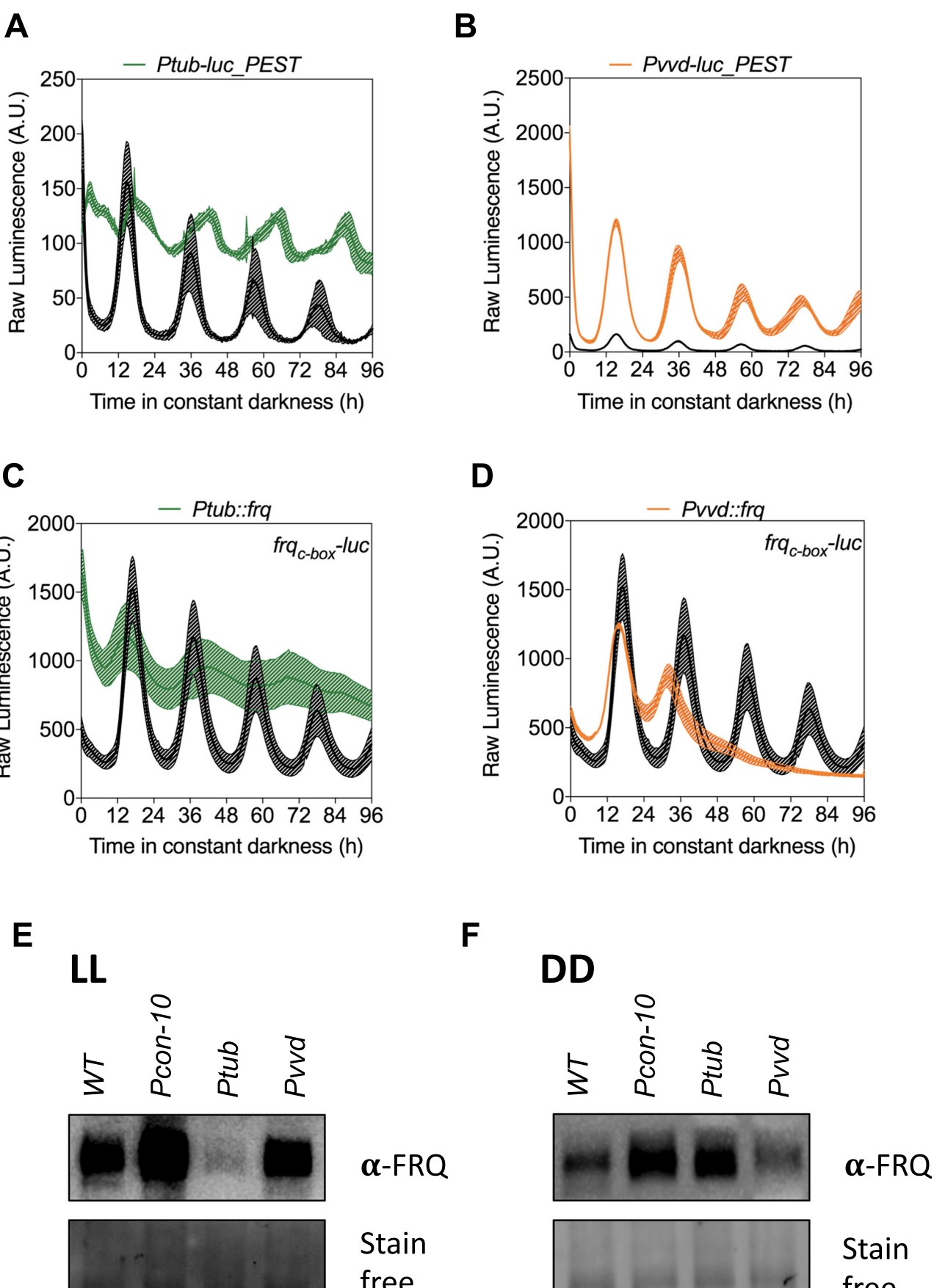

◀ **Figure EV2.  Hybrid oscillators with *tubulin* and *vivid* promoters exhibit weak rhythms after three days in LD 12:12 entrainment.**

(**A,B**) Evaluation of *tubulin* (green (**A**)) and *vivid* (orange (**B**)) *promoters* under DD conditions. The strains were grown for three days under 12:12 LD cycles prior start recording in darkness. The black traces represent a $frq_{c-box+pLRE}$-*luc_PEST* reporter. In all cases, experiments were run three independent times, and a representative set is shown. Each luciferase trace corresponds to the average of three different wells ± SD. (**C,D**) Evaluation of HOs under DD conditions, by analyzing LUC activity coming from a $frq_{c-box}$-*luc* reporter. The black traces represent a *wild type* strain, whereas the different HOs are depicted in color. In green the *tubulin promoter* (**C**) and in orange the *vivid promoter* controlling *frq* transcription (**D**). In all cases, experiments were run three independent times, and a representative set is shown. Each luciferase trace corresponds to the average of three different wells ± SD. (**E,F**) Western blots showing the levels of FRQ in the different HOs, after growth in LL for 48 h (**A**) and after 24 h in DD (coming from 24 h in LL) (**B**). The name of each sample indicates the promoter that controls *frq* transcription.

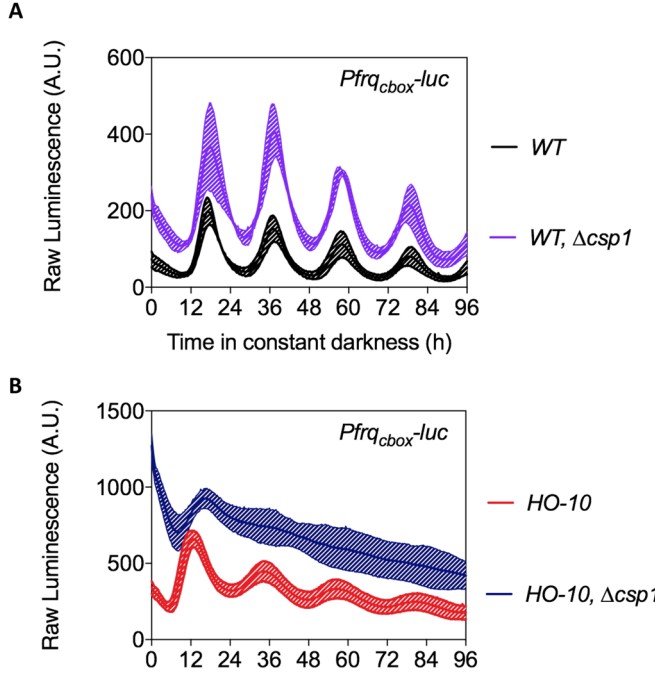

**Figure EV3. Effect of the absence of *csp-1* in the WT and HO-10 oscillators.**

(A,B) Evaluation of the WT (A) and HO-10 oscillator (B) in the absence of *csp-1*. Strains were entrained for 3 days under 12:12 LD cycles prior to monitoring in DD. Strains contain a *c-box-luc* reporter. In (A), the black traces represent the WT strain and purple traces the Δ*csp-1* in a WT clock background. In (B), the red traces represent the HO-10 and blue traces the Δ*csp-1* in a HO-10 background. In all cases, experiments were run three independent times, and a representative set is shown. Each luciferase trace corresponds to the average of three different wells ± SD.

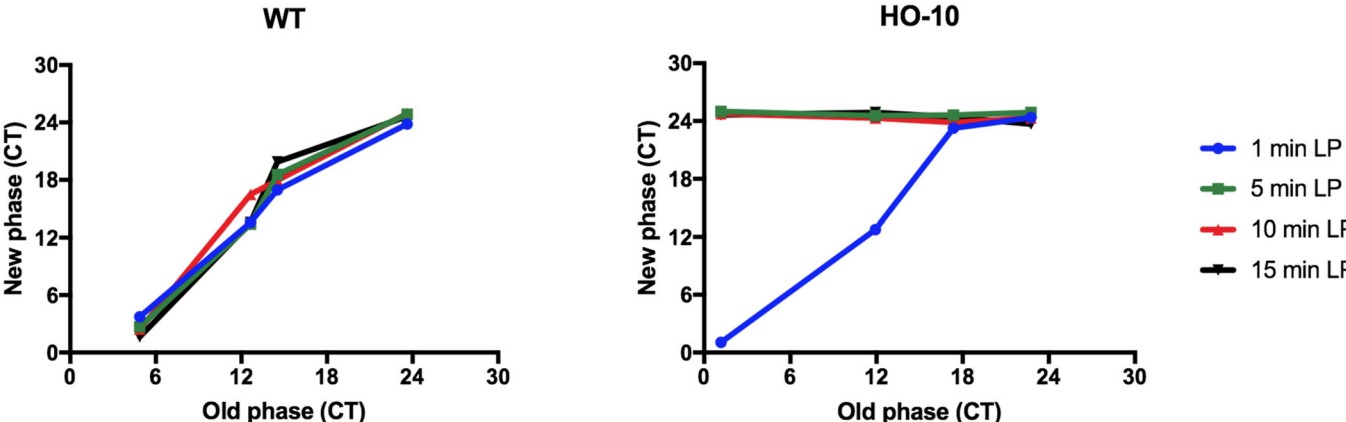

**Figure EV4.   HO-10 mainly exhibits type 0 PTCs.**

The phase shift produced by discrete light pulses (LPs) of different duration was evaluated in a *WT* (left panel) and *HO-10* (right panel) clock. Strains were entrained as in Fig. 4G,H and after 48 h in DD and LP of 1, 5 10, and 15 min were administrated. While for all different tested LPs the WT clock exhibited a Type 1 PTC, the HO-10 depicted a type 0 PTC for LPs of 5, 10, and 15 min. Only for the shorter LP (of 1 min) a Type 1 PTC was obtained. The reported phase shifts were calculated by comparing the new phase after LP to the phase of the same strain without LP. *frq*<sub>*c-box+pLRE*</sub>*–luc-PEST* was used as reporter. In all cases, experiments were run three independent times, and a representative set is shown. Each luciferase trace corresponds to the average of three different wells. Standard deviation is not plotted to help visualization of each independent PTC.

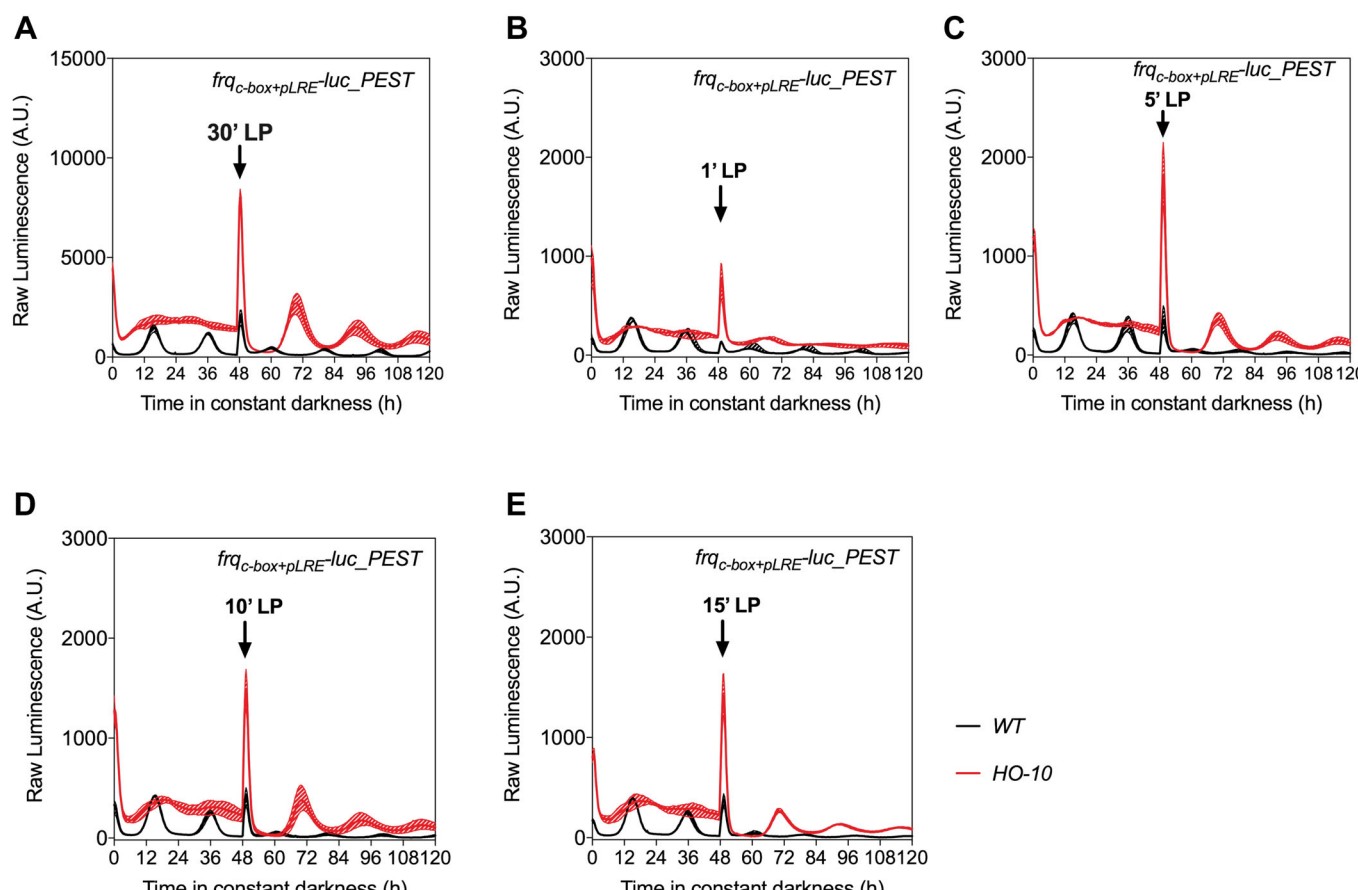

**Figure EV5. A 5-min LP is sufficient to trigger oscillations in HO-10.**

(**A–E**) After identifying that a 30 min LP could "jump start" the HO-10 (**A**), we evaluated the effect of different duration LPs in *WT* (black) and *HO-10* (red) strains. Strains were transfer from 24 h LL and start monitoring LUC activity in DD, after 48 h a LP of 1 (**B**), 5 (**C**), 10 (**D**), 15 (**E**) min was administrated. *frq*<sub>c-box+pLRE</sub>-*luc-PEST* was used as a reporter. In all cases, experiments were run three independent times, and a representative set is shown. Each luciferase trace corresponds to the average of three different wells ± SD.

