## [Peer Review File · The EMBO Journal]

Transcriptional rewiring of an evolutionarily conserved circadian cloc

Alejandra Goity, Andrey Dovzhenok, Sookkyung Lim, Christian Hong, Jennifer Loros, Jay Dunlap, and Luis Larrondo

Corresponding author: Luis Larrondo (llarrondo@bio.puc.cl)

Review Timeline:

Submission Date:	30th Jun 23
Editorial Decision:	9th Aug 23
Revision Received:	22nd Dec 23
Editorial Decision:	29th Jan 24
Revision Received:	5th Mar 24
Accepted:	8th Mar 24

Editor: Ieva Gailite

Transaction Report:

Dear Dr. Larrondo,

Thank you for submitting your manuscript for consideration by the EMBO Journal. We have now received comments from three reviewers, which are included below for your information.

As you will see from the reports, all reviewers find the study per se of interest, while also pointing out a number of important aspects that would need to be addressed in the final manuscript before they can recommend acceptance of the manuscript. Based on the interest expressed in the reports, I would like to invite you to address the issues raised by the referees in a revised manuscript. In particular, experimentally addressing the points raised by reviewers #1 and #3 regarding the potential alternative explanations of the presented findings would be important for acceptance here. I think it would be useful to discuss the revision in more detail via email or phone/videoconferencing - please let me know which option you prefer.

We generally allow three months as standard revision time. As a matter of policy, competing manuscripts published during this period will not negatively impact on our assessment of the conceptual advance presented by your study. However, please contact me as soon as possible upon publication of any related work to discuss the appropriate course of action. Should you foresee a problem in meeting this three-month deadline, please contact us to arrange an extension.

When preparing your letter of response to the referees' comments, please bear in mind that this will form part of the Review Process File and will therefore be available online to the community. For more details on our Transparent Editorial Process, please visit our website: <https://www.embopress.org/page/journal/14602075/authorguide#transparentprocess>. Please also see the attached instructions for further guidelines on preparation of the revised manuscript.

Please feel free to contact me if you have any further questions regarding the revision. Thank you for the opportunity to consider your work for publication. I look forward to discussing your revision.

With best regards,

leva

leva Gailite, PhD
Senior Scientific Editor
The EMBO Journal
Meyershofstrasse 1
D-69117 Heidelberg
Tel: +4962218891309
i.gailite@embojournal.org

We realize that it is difficult to revise to a specific deadline. In the interest of protecting the conceptual advance provided by the work, we recommend a revision within 3 months (7th Nov 2023). Please discuss the revision progress ahead of this time with the editor if you require more time to complete the revisions.

Referee #1:

In this work Goity et al investigate the properties of a *Neurospora crassa* circadian clock where the promoter of core negative clock regulator frequency was replaced by that of clock-controlled output gene *con-10*. The resulting "hybrid oscillator" HO-10 exhibits an oscillating temperature compensated clock with a longer period compared to the wildtype that requires prior entrainment for a robust oscillation in constant darkness. Peak levels of FRQ expression in constant light are ca. 1.7x and after 24h in constant darkness ca. 3x higher in HO-10 than wildtype. Interestingly, the peak of conidiation occurs at a fixed time after (or before) lights on, as opposed to the wildtype where the reference cue is the light to dark transition.

The authors provided thorough analysis of their HO-10 strain, however, I have one major issue with the manuscript: The authors base their interpretation of the results on the assumption that the *con-10* promoter is not directly controlled by the core clock transcription factor WCC but indirectly via an unspecified WCC-activated transcription factor. They base this assumption on two studies where in ChIPseq analyses with WCC subunit WC-2 no binding sites were mapped to the *con-10* promoter. However, this is a classical absence of evidence and does not prove at all that WCC is not directly controlling *con-10*. In fact, there is more compelling evidence that *con-10* is likely a direct target of the WCC:

1. *con-10* is rapidly/immediately light inducible to high levels with an equally rapid light adaptation similar to e.g. *frq* and *vvd* (Fig. 4H in this manuscript and Tan et al 2004).
2. *con-10* has been identified as an oscillating cgc in independent deep sequencing analyses (RNA-seq, PolIII ChIP).
3. The *con-10* promoter fragment cloned in front of the *frq* ORF is peppered with GATC/G repeats, some of them with similar spacing as in the *frq* LRE as well as *c-box*.

The authors have to investigate experimentally (e.g. by ChIP PCR) if the WCC binds to the wt *con-10* promoter as well as the promoter of the HO-10 hybrid *frq* gene in a light dependent or circadian manner or not. Much depends on the outcome of this experiment and potentially the whole manuscript would have to be rewritten accordingly.

Due to the extremely rapid light adaptation of *con-10* the ChIP PCR should be done 5' after lights on using multiple sets of primers that cover the whole cloned promoter fragment.

Other comments:

- The observed phenotypes of HO-10 could be also be caused by the increased FRQ protein levels and its altered/lower peak to trough ratio. This may also explain why entrainment is required in order for HO-10 to oscillate in the ensuing free run but not anymore in HO-10 Δvvd double mutant, as Δvvd leads to increased FRQ levels in the light with marginal or no changes of FRQ trough levels in constant darkness. Elevated FRQ levels may also explain the longer period length. It would be interesting to then investigate if this may also be the cause for lights-on timer logic.
- 251ff. It is interesting that on a molecular level the core clock continues to oscillate at all tested temperatures despite a *frq* mRNA lacking the *frq* 5'UTR. Considering that the *frq* 5'UTR mutants were analyzed with racetube assays a few years before the luciferase reporter assay was established for *Neurospora* clock research, it seems as if deletion of the 5'UTR imbalances the system such that the output is impaired while the core clock still runs, once again showing that the core clock oscillator seems to be more resistant to alterations in protein levels of the core clock machinery than the output.
- 370ff. "the rate of *frq* transcription is significantly reduced in LL..." Yet, FRQ levels in LL are higher in HO-10 than in wt as you show in the expanded view figures. Light induced transcription of *con-10*, according to Tan et al, appears to be almost an order of magnitude higher than *frq* and adapted levels may not be so different considering your observations.
- 419ff. promoters of *wc-1* and *frq* contain CSP-1 binding sites. Thus, the observation that the "behavior of $\Delta csp-1$ resembles HO-10" could also be explained by the increased overall FRQ levels and/or a change in its peak/trough ratio, which might be

worthwhile to investigate.

Referee #2:

In this manuscript, Goity and co-authors present a study on a semi-synthetic gene circuit of circadian clock system in *Neurospora*. While the transcriptional-translational feedback loop (TTFL) is evolutionarily conserved and widely adopted in most clock systems, there is a remaining question whether the one-step TTFL is the sole way for transcriptional clock systems. The newly engineered hybrid oscillator (HO-10) exhibits circadian properties resembling those of the native clock system, suggesting that the canonical TTFL may not be the exclusive pathway for generating cycles. Interestingly, the author found that HO-10 displays a differential property in light adaptation ("lights-on timer"). The obtained results are clear and well-analyzed from the perspective of chronobiology. This reviewer would like to suggest the authors clarify a few points to improve the manuscript.

Comments:

- 1) This is interesting study using *Neurospora* as a model system. If authors mention benefits of using *Neurospora* in synthetic biology, it may help readers to understand why authors choose fungi in this study.
- 2) In HO-10 case, is the site recognized by TF (Fig.1A) totally specific to TF?
As this is the "semi-synthetic" transcriptional circuit, the reviewer wonders the possibility that other TFs may also affect the transcriptional rhythms. How do authors exclude this possibility? Also, authors state that "the description of HO-10 transcriptional control, and exact circuit topology, will be described elsewhere" in Results (line 172 - 173). The reviewer understands that details of HO-10 transcriptional control and topology are out-of-focus of the paper, but, as HO-10 is the main material, the authors are recommended to add more details of HO-10 in the texts.
- 3) Their data suggests that the period length of the semi-synthetic loop-based oscillator (HO-10) is longer than WT and peaks slightly earlier than WT (Fig.2). Is this the general property of this "hybrid oscillator" or just specific to HO-10? This reviewer appreciates if authors discuss this point in Discussion.
- 4) The reviewer believes Fig.5E supports simulation data that they showed in (D). The reviewer appreciates if authors quantitatively and statistically analyze data (E) so that readers easily compare D and E.
- 5) The circadian rhythm changes many physiological aspects, and environmental factors also affect circadian machinery. Did author observe any physiological change of HO-10 such as the metabolic state and proliferation?
- 6) A part of labels at Fig.4H is out of sight.
- 7) There is no explanation for the asterisk and the error bars in Fig.EV4.

Referee #3:

The authors take on a fascinating question that is fundamental to the field of circadian biology--whether the native and highly conserved architecture of transcriptional regulation is required for rhythmic function. The manuscript reports the creation of a semi-synthetic oscillatory strain HO-10 where *frq* is placed under the control of the *con-10* promoter, known to be a clock-controlled promoter in constant darkness in the wildtype.

The strongest conclusions of the manuscript, in my view, are that the amplitude and entrained phase of the resulting HO-10 oscillator are plastic and depend presumably on specific regulatory elements in the *con-10* promoter. By contrast, the period of oscillation and temperature compensation properties are highly robust to this "rewiring" as nicely shown by the mutational study presented in Fig 2.

However, I was left wondering how strong the evidence is that TTFL regulation is absolutely required for free-running rhythms in this system. An alternative hypothesis might be that the *con-10* promoter works not because it is rhythmic but because it gets close to matching the WT expression level of *FREQ* in DD (Fig EV4 suggests that the other semisynthetic HO systems had markedly higher levels of *FREQ*). The clearest evidence that rhythmic expression of *FREQ* is required would be to show that a non-rhythmic promoter with similar strength to *Pcon-10* gives completely arrhythmic output in molecular assays. If the tools do not exist to do this experiment, I could certainly be convinced that it is beyond the scope of this study.

other comments:

* "Temperature compensation is unlikely a network property"--meaning is somewhat unclear. Do the authors specifically mean "transcriptional network"?

*I find figure 5B somewhat confusing. The bifurcation diagram in black seems to imply that the rate constant describing activity of the con-10 promoter is treated as a parameter of the model (on the horizontal axis) and a Hopf bifurcation occurs at a critical value of this rate constant. But then the red arrow describing the effect of a discrete light pulse seems to suggest that the activity of the parameter is permanently changed by the light pulse pushing the system into the oscillatory region. This suggests that the "parameter" k_{24} is actually a dynamical variable in the model that can remember prior stimulation by a light pulse. Please clarify how we should read this figure and how the light pulse treatment is handled in the model.

We sincerely thank all three reviewers for their thoughtful input and expert comments. We have incorporated their suggestions, as well as additional data that have strengthened the main conclusions of the work. At the same time, we have stated the limitations of our study, while highlighting its relevance. We consider that this revised submission is a better version of our MS, a view that we hope to transmit in this rebuttal letter and in the main text as well.

Line numbers, indicating the modified text, is based on the new MS.pdf main file. The .doc file, with track changes, has been also uploaded.

Referee #1:

In this work Goity et al investigate the properties of a *Neurospora crassa* circadian clock where the promoter of core negative clock regulator frequency was replaced by that of clock-controlled output gene *con-10*. The resulting "hybrid oscillator" HO-10 exhibits an oscillating temperature compensated clock with a longer period compared to the wildtype that requires prior entrainment for a robust oscillation in constant darkness. Peak levels of FRQ expression in constant light are ca. 1.7x and after 24h in constant darkness ca. 3x higher in HO-10 than wildtype. Interestingly, the peak of conidiation occurs at a fixed time after (or before) lights on, as opposed to the wildtype where the reference cue is the light to dark transition.

The authors provided thorough analysis of their HO-10 strain, however, I have one major issue with the manuscript: The authors base their interpretation of the results on the assumption that the *con-10* promoter is not directly controlled by the core clock transcription factor WCC but indirectly via an unspecified WCC-activated transcription factor. They base this assumption on two studies where in ChIPseq analyses with WCC subunit WC-2 no binding sites were mapped to the *con-10* promoter. However, this is a classical absence of evidence and does not prove at all that WCC is not directly controlling *con-10*. In fact, there is more compelling evidence that *con-10* is likely a direct target of the WCC:

1) *con-10* is rapidly/immediately light inducible to high levels with an equally rapid light adaptation similar to e.g. *frq* and *vvd* (Fig. 4H in this manuscript and Tan et al 2004).

We thank the reviewer for this insightful analysis and comment, as he/she raises a valid and relevant point. We agree that absence of evidence is not evidence of absence. Nevertheless, it is important to highlight that -so far- all the published ChIP-Seq studies point to a similar conclusion (no obvious WCC binding at the *con-10* promoter).

We have now analyzed an additional set of unpublished ChIP-Seq for WC-2, focusing on a light pulse (15 min). The data (see Figure Reb2, in point 3) depicts the IGV for NCU02265 (*frq*) and NCU07325 (*con-10*) both to the same scale: no evident WC-2 binding is seen in *con-10*, whereas *frq* shows two distinct peaks matching the pLRE region: in its promoter as well as in the antisense promoter (*qrf*).

We also agree with the reviewer that *con-10* has hallmarks of a light induced gene, yet its expression kinetics are different from *frq* and *vvd*. Thus, RNA-Seq data (Wu et al 2014) shows that in response to light treatments of 15, 60 and 120 min *con-10* has maximum response after 60 min, whereas *frq* and *vvd* have their peaks only after 15 min (Table 1).'

Table 1. Data extracted from Wu et al. 2014. In yellow it is highlighted the peak of each gene.

Locus	Symbol	Name	DD-L15m Ratio	DD-L60 Ratio	DD-L120 Ratio
NCU02265	frq	frequency	2.7	2.4	2.3
NCU07325	con-10	conidiation-10	4.8	7.1	4.3
NCU03967	vvd	vivid	6.6	5.1	4.0

Likewise, *con-10* delayed peak of expression (relative to *frq* or *vvd*) agrees with previous observations (Chen et al. 2009) (Figure Reb1), in which late light response genes were defined based on having their peak of expression at 30 min or later.

Figure Reb1. Light response of NCU02265(*frq*), NCU07325(*con-10*) and NCU03967 (*vvd*) evaluated by microarrays. Measurements were performed after 5, 10, 15, 30, 45, 60, 120 and 240 minutes of light exposure. Replotted data extracted from Chen et al. 2009.

Importantly, in the text **we have now explicitly acknowledged** the fact that although existing datasets fail to confirm WCC binding to the *con-10* promoter, such negative result cannot fully rule out that there is no binding. Nevertheless, even in a scenario where transient WCC binding could be eventually confirmed, this would not affect the main point of the work (an altered yet functional topology of a TTFL) nor the main interpretations/conclusions of the MS. To better clarify this, and to strengthen the main message of the work, we have added additional data that highlight how in HO-10 *frq* expression is under distinct transcriptional control, different to a WT oscillator. In the MS, **new Figure EV6** shows how, as predicted, a TF known to be part of output pathway (but not essential for the clockworks), has now a pivotal role in HO-10 rhythms. Thus, CSP-1, which is controlled by WCC, is now by definition a core-clock component of this extended TTFL topology, since when deleted in HO-10, rhythms are completely abrogated, whereas absence of *csp-1* does not affect WT clock rhythmicity (Sancar et al., 2012).

We have modified the text accordingly, and changed the wording of different parts of the text as listed herein:

- Line 101: "Nevertheless, neither its induction by light nor its rhythmic expression have been reported to be directly controlled by the WCC (Hurley et al, 2014; Smith et al, 2010), and instead it has been described to depend on a complex regulation involving other transcription factors."

Line 190: "As a proof of concept that HO-10 differs in its molecular circuitry from a WT clock, we assessed the consequences of eliminating an output regulator. We chose a TF known to be under clock control and that is part of the output pathways, but that it is not essential for clock function. Therefore, we deleted *csp-1*, which encodes for a TF that modulates a large number of ccgs, including *con-10*. In a WT clock devoid of *csp-1*, rhythms remain robust (Fig EV6A) and period is unaltered under low sugar conditions (Sancar et al, 2012). In contrast, deletion of *csp-1* in HO-10 causes arrhythmicity (Fig EV6B) confirming that - by definition- this output transcription factor is now a bona fide core-clock component and, therefore, part of the extended TTFL topology of HO-10. "

Line 256: "...the WCC and its binding to the pLRE in the *frq* promoter, which leads to defined transcriptional dynamics (Froehlich et al, 2002; Oehler et al, 2023). Importantly, this region is no longer controlling *frq* expression in HO-10 and, moreover, a classic activation via WCC (at least with strong promoter binding) is not occurring in the *con-10* promoter (Smith et al, 2010; Hurley et al, 2014; Sancar et al, 2015a); therefore, in HO-10 light information is conveyed to *frq* by additional steps, involving other regulators downstream from WCC activity"

Line 374: "In this new architecture the canonical c-box and pLRE sequences (key in regulating native *frq*) are no longer controlling its expression and importantly, *frq* transcription (commanded now by the *con-10* promoter) ceases to be simply and directly regulated by the WCC. Although multiple ChIP-Seq datasets, including our own unpublished ones, have failed to detect WCC occupancy at the *con-10* promoter either under DD conditions or after a light-pulse (Smith et al, 2010; Hurley et al, 2014; Sancar et al, 2015a), we cannot fully discard a transient action of WCC on the *con-10* promoter (i.e. WCC acting as a pioneer TF facilitating the recruitment of additional components). Nevertheless, it is clear that the rewired TTFL presents dynamics that are quite different from the canonical *frq* promoter (given by its native elements), which exhibits distinct refractoriness or complex transcriptional and chromatin remodeling dynamics (Oehler et al, 2023; Cesbron et al, 2015). Future efforts will be focused on providing a detailed transcriptional landscape of this rewired oscillator, including the main TFs acting as bona fide positive elements. As shown in Fig EV6, deletion of a TF known to be part of the output pathways, such as CSP-1, confirms that although this TF is not essential for a WT clock, its absence causes arrhythmicity in the context of HO-10: in other words, CSP-1 is now, by definition, a genuine core-clock component. Although FRQ can still directly inhibit WCC activity, the latter now also controls transcription of *frq* via output components that act upon the *con-10* promoter, such as CSP-1, and likely many clock-controlled TFs such as SUB-1 (Sancar et al, 2015a)".

2) *con-10* has been identified as an oscillating ccg in independent deep sequencing analyses (RNA-seq, PolII ChIP).

Indeed, while *con-10* has been identified as a ccg (Hurley et al 2014) and a light responsive gene (Corrochano et al. 1995) it is also true that not all ccgs and light response genes are strictly WCC targets (Chen et al. 2009, Sancar et al., 2015).

The modified text (see examples above) recognizes that even if WCC plays some role in *con-10* expression, HO-10 has a complex and distinct transcriptional regulation, which is clearly exemplified with the absence of HO-10 rhythms when *msp-1* is absent (**Fig EV6**). A future publication will focus on the exact topology of the extended TTFL, characterizing the different TFs that are part of the HO-10 circuitry, and where the exact contribution of WCC will be fully assessed.

3) The *con-10* promoter fragment cloned in front of the *frq* ORF is peppered with GATC/G repeats, some of them with similar spacing as in the *frq* LRE as well as c-box.

Indeed, there are several GATC/G in the promoter area, with various of them partially overlapping with experimental sites for other TFs (Sancar et al., 2015). It is also important to remember that throughout the genome not all predicted GATC/G sequences may be active binding sites, and among the latter some may be "functional" only under particular conditions. As commented earlier, in three published WC-2 ChIP-Seq analyses (plus another unpublished one, see below), WC-2 (WCC) association to *con-10* (NCU07325) has not been observed:

- Smith et al. 2010: WCC direct targets in response to a 8-min light pulse, with no binding at the *con-10* locus.

- Hurley et al. 2014: WCC circadian targets. In this particular case, not only a single time point, but a full time course (From DD4 to DD32 h) also failed to reveal WC-2 occupation at the *con-10* promoter.
- Sancar et al. 2015: In this work two ChIP-Seq approaches were performed. In the first one, chromatin was fragmented by sonication and WCC binding sites were identified by tandem affinity ChIP of a TAP-tagged WC-2. The second approach, the chromatin was subjected to MNase digestion and ChIP was performed using a WC-2 antibody. None of these strategies identified binding of WCC to the *con-10* promoter.

In addition, and as mentioned above, our own unpublished ChIP-Seq of WC-2 after 15 minutes of light exposure failed to detect the binding of WCC in *con-10* promoter (Figure Reb2).

Figure Reb2. Screen shots from IGV of regions surrounding NCU02265 (*frq*) at the left and NCU07325 (*con-10*) at the right. Both are at the same scale. WC-2 ChIP-Seq after 15 min of light.

Other transcription factors, such as SUB-1 and CSP-1 have been identified as *con-10* regulators, including their confirmed binding to the *con-10* promoter (Sancar et al 2015 PMC4378982, Sancar et al 2015 PMC4381671, Sancar et al 2011 PMC22152473). It is important to comment that SUB-1 is a GATA family TF, just like WC-1 and WC-2, with the potential to recognize promoter sequences with certain similarity to the ones of WCC. Moreover, the Brunner's lab tool NEUTRA allows the easy visualization of the binding sites of WCC, CSP-1, FF7 and SUB-1 identified by ChIP-Seq experiments. Analysis of the *con-10* promoter region reveals the binding of CSP-1, FF7 and SUB-1, yet, the browser does not depict WCC binding (Figure Reb3). At the same time in *frq* (NCU02265) the binding of the four transcriptions factors, including WCC can be readily seen (Figure Reb4). Notably, deletion of any of them (except for WCC) has little effect on rhythmic *frq* expression and the clockworks.

Figure Reb3. Screenshot of NEUTRA tool showing the binding sites of CSP1, FF7, WCC and SUB1 in *con-10* (NCU07325) promoter region. (https://neutra.bzh.uni-heidelberg.de/jbrowse/JBrowse-1.12.3/index.html?loc=4%3A4364026..4368240&tracks=DNA%2CCSP1_BindingSites%2CFF7_BindingSites%2CWCC_BindingSites%2CSUB1_BindingSites%2CNC10&highlight=)

Figure Reb4. Screenshot of NEUTRA tool showing the binding sites of CSP1, FF7, WCC and SUB1 in *frq* (NCU02265) promoter region. (https://neutra.bzh.uni-heidelberg.de/jbrowse/JBrowse-1.12.3/index.html?loc=7%3A1115761..1132400&tracks=DNA%2CCSP1_BindingSites%2CFF7_BindingSites%2CWCC_BindingSites%2CSUB1_BindingSites%2CNC10&highlight=)

In toto, while our current experimental data does not allow us to fully rule out any participation of WCC in *con-10* expression, it is evident that the *con-10* promoter bears a distinctive regulatory complexity compared to *bona fide* WCC targets such as *vvd* and *frq*. For example (Figure Reb5), in response to light *frq* or *vvd* are rather unaffected by the absence of *sub-1* or *ff-7* whereas, on the other hand, *con-10* expression is severely compromised (~50%) when such TFs are absent (Sancar et al. 2015). This argues in favor that light response of *frq* and *vvd* are largely commanded by the WCC, whereas in the case of *con-10* other transcription factors (such as SUB-1) play relevant roles.

Figure Reb5. Comparison of *frq*, *vvd* and *con-10* expression patterns. The graphs, generated from data extracted from Sancar et al. 2015, depict expression of the corresponding genes in the absence of *sub1* and *ff7*. RNA-seq data was generated from samples kept in DD or subjected to light for 30, 60 and 120 minutes

4) The authors have to investigate experimentally (e.g. by ChIP PCR) if the WCC binds to the wt *con-10* promoter as well as the promoter of the HO-10 hybrid *frq* gene in a light dependent or circadian manner or not. Much depends on the outcome of this experiment and potentially the whole manuscript would have to be rewritten accordingly.

Due to the extremely rapid light adaptation of *con-10* the ChIP PCR should be done 5' after lights on using multiple sets of primers that cover the whole cloned promoter fragment.

We considered performing ChIP-PCR for WC-2 on *con-10*, as suggested by the Reviewer. Yet, even if we would have found no binding (as already reported in publications from various labs), the initial caveat raised by the reviewer would have still remained: "absence of evidence is not evidence of absence". Moreover, since we and others will continue to work on HO-10 and/or WCC-regulation, we are aware that some weak binding of WCC to *con-10* may be (or not) eventually detected, under particular conditions. Therefore, we explicitly recognize that although WCC binding has not been confirmed, it may still play a role in regulating the *con-10* promoter. **Nevertheless, the relevant point to keep in mind is that even if it does, the control of *frq* expression in HO-10 is quite different from the classic WCC control exerted on the *frq* promoter, and that moreover, several other TFs are playing essential roles in HO-10 (i.e. CSP-1), whereas in a WT oscillator WCC is otherwise the main essential circadian TF.**

We surmise that the biggest concern of the reviewer may be the fact that reports such as Sancar et al. 2015 conducted a ChIP-Seq of WC2 and didn't identify some targets of WCC by this technic, and instead they were able to detect binding by ChIP-PCR in some tested targets. When we looked the reported light response of some of those genes, we noticed that none of them (with the exception of NCU08769) had diminished responses in the absence of *sub-1* or *ff-7*, which contrasts to what is seen in *con-10* (NCU07325) (Figure Reb6). Thus, the confirmed ChIP-PCR WCC targets appear more similar to *frq* and *vvd* (in terms of their independence of other TFs), yet with a weaker light response than *frq* or *vvd*. Although NCU08769 shows strong light responses, it is positively affected by the absence of *sub-1* (the opposite than *con-10*).

Figure Reb6. Light response of WCC targets identified by ChIP-PCR of WC-2 and *con-10*. The light response was measured in dark and after 30, 60 and 120 minutes in light. The response was evaluated in WT, $\Delta sub1$ and $\Delta ff7$. Data were extracted from Sancar et al. 2015.

Also, when we analyzed these genes regarding the fold change to light, *con-10* has a higher fold change and with a strong responses after 60 and 120 minutes (Figure Reb7).

Figure Reb7. Fold change in light response after 30, 60 and 120 minutes of light. Data extracted from Sancar et al. 2015.

Finally, to compare the light-triggered transcription of NCU07325 (*con-10*) in comparison to NCU02265 (*frq*) and NCU03967 (*vvd*) we revisited the published ChIP-Seq data of PolII (ser5-P, ser2-P and total polII) after a 5 min Light-Pulse (Cesbron et al. 2015). The fold change after the first light pulse is ~20X for *frq* and *vvd*, and for *con-10* ~2.5X (Figure Reb8)

Once again, while this evidence does not discard the possibility that WCC binds the *con-10* promoter, it highlights that *con-10* transcription kinetics are quite different from confirmed “weak” (Fig Reb 6, 7), or “strong” (Fig Reb8) WCC targets

Figure Reb8. Fold change in temporal profile in ser5-5P (left), ser-2P (middle) and total polII (right) for NCU07325, NCU02265 and NCU03967 after light pulses. Data extracted from (Cesbron et al. 2015).

We considered that a potential caveat of our rationale, is that all of the above-mentioned ChIP-Seq experiments were conducted in liquid cultures samples, whereas our luciferase studies (tracking the HO-10 oscillator or *con-10*) are conducted in solid-agar media. Therefore, to confirm that our conditions (solid media) are comparable to the published ones (liquid media), we decided to contrast *con-10* responses to light in liquid and solid media after a 10 min light pulse. After the LP, we measured LUC activity every 5 minutes in constant darkness using a *Pcon10-luc^{PEST}* reporter and a CCD camera. When we analyzed the results, we observed no obvious differences in solid vs liquid (Figure Reb9), with max response at 1.2 and 1.0 h respectively. We performed three independent times the experiment and there is no difference between the time it takes to reach the max response (h)

in solid vs liquid. This suggest that the responses we observed in liquid and solid are mechanistically similar.

Figure Reb9. Liquid: Strains grew for ~24 hours in petri dishes with VM 2% sucrose in LL, after the mycelia mat was formed (and before conidiation started), the discs were cut, washed and transferred to petri dishes with liquid LNN-CCD + QA. **Solid:** Conidia suspensions were used to inoculated 96-well plate with solid LNN-CCD + QA. Strains were for 24 hours in LL and then transferred to DD. Samples were in DD for 18 hours and a 10 min light pulse was given. *Pcon-10-luc^{PEST}* activity were monitored in DD using a CCD camera with a 5 minutes acquisitions.

Other comments:

5) - The observed phenotypes of HO-10 could be also be caused by the increased FRQ protein levels and its altered/lower peak to trough ratio. This may also explain why entrainment is required in order for HO-10 to oscillate in the ensuing free run but not anymore in HO-10 Δvvd double mutant, as Δvvd leads to increased FRQ levels in the light with marginal or no changes of FRQ trough levels in constant darkness. Elevated FRQ levels may also explain the longer period length. It would be interesting to then investigate if this may also be the cause for lights-on timer logic.

The reviewer provides an interesting interpretation. Our current working model considers that it is not just about absolute quantities of the state variable FRQ, but also about its quality (i.e. timing of the predominant phospho-forms). Regarding the latter, the peak/trough of *frq* expression may play a key role in eliciting effective windows of repression and no-repression. Recent and future mechanistic studies will continue on providing the nuts and bolts of the transcriptional aspects of clock dynamics.

6) - 251ff. It is interesting that on a molecular level the core clock continues to oscillate at all tested temperatures despite a *frq* mRNA lacking the *frq* 5'UTR. Considering that the *frq* 5'UTR mutants were analyzed with racetube assays a few years before the luciferase reporter assay was established for *Neurospora* clock research, it seems as if deletion of the 5'UTR imbalances the system such that the output is impaired while the core clock still runs, once again showing that the core clock oscillator seems to be more resistant to alterations in protein levels of the core clock machinery than the output.

Indeed, we are also excited with these results. We have added a reflection along these lines

Line 427: "Another interesting observation is that HO-10 exhibits robust molecular oscillations, whereas overt conidiation rhythms are impaired, as seen also in some other mutants (Larrondo et al, 2015; Shi et al, 2007). Thus, the core-oscillator appears more resilient to changes in state variables compared to the clock output, which could be interpreted as that the oscillator cogwheels need to be of a given size (i.e. amplitude) to properly engage with the cogs controlling overt output. "

7) - 370ff. "the rate of *frq* transcription is significantly reduced in LL..." Yet, FRQ

levels in LL are higher in HO-10 than in wt as you show in the expanded view figures. Light induced transcription of con-10, according to Tan et al, appears to be almost an order of magnitude higher than frq and adapted levels may not be so different considering your observations.

Thanks for the point. We have reflected on that

Line 432: *"While we have interpreted our results based on relative FRQ levels, we are not oblivious to the fact that quality of FRQ (i.e. timing on phospho-forms) may be, potentially, playing even stronger effects"*

8) - 419ff. promoters of wc-1 and frq contain CSP-1 binding sites. Thus, the observation that the "behavior of Δ csp-1 resembles HO-10" could also be explained by the increased overall FRQ levels and/or a change in its peak/trough ratio, which might be worthwhile to investigate.

That is a good point. We have commented this now in the text.

Line 357: *"Such role could be the sum of CSP-1 effects on the wc-1 and frq promoters, where the absence of csp-1 is evidenced by an altered phase in DD following distinct entrainment conditions."*

Referee #2:

In this manuscript, Goity and co-authors present a study on a semi-synthetic gene circuit of circadian clock system in *Neurospora*. While the transcriptional-translational feedback loop (TTFL) is evolutionarily conserved and widely adopted in most clock systems, there is a remaining question whether the one-step TTFL is the sole way for transcriptional clock systems. The newly engineered hybrid oscillator (HO-10) exhibits circadian properties resembling those of the native clock system, suggesting that the canonical TTFL may not be the exclusive pathway for generating cycles. Interestingly, the author found that HO-10 displays a differential property in light adaptation ("lights-on timer"). The obtained results are clear and well-analyzed from the perspective of chronobiology. This reviewer would like to suggest the authors clarify a few points to improve the manuscript.

Comments:

1) This is interesting study using *Neurospora* as a model system. If authors mention benefits of using *Neurospora* in synthetic biology, it may help readers to understand why authors choose fungi in this study.

Thanks for bringing this up. A phrase has been included now in the text.

Line 116: *"Due to the abundance of molecular tools, straightforward genetics, and the absence of gene families and paralogues, Neurospora is a great platform in which to adopt synthetic biology strategies, such as the implementation of new circadian clock topologies or synthetic circuits (Tabilo-Agurto et al, 2023; Matsu-Ura et al, 2018)"*

2) In HO-10 case, is the site recognized by TF (Fig.1A) totally specific to TF? As this is the "semi-synthetic" transcriptional circuit, the reviewer wonders the possibility that other TFs may also affect the transcriptional rhythms. How do authors exclude this possibility? Also, authors state that "the description of HO-10 transcriptional control, and exact circuit topology, will be described elsewhere" in

Results (line 172 - 173). The reviewer understands that details of HO-10 transcriptional control and topology are out-of-focus of the paper, but, as HO-10 is the main material, the authors are recommended to add more details of HO-10 in the texts.

We appreciate the reviewer understanding that the detailed topology of HO-10 is beyond the scope of the current MS. Indeed, the illustration in Fig 1A aims to depict the overall logic of the strategy. Undoubtedly, the regulation of HO-10 promoter is the consequence of many transcription factors, of which we have now included one explicit example, CSP-1 (as seen in new Fig SV6). Also, now in Fig 1A TF has been replaced by TFs.

We have also expanded the discussion on how HO-10 transcriptional dynamics differ from the WT system (see for example new paragraphs in lines 190, 258 or 374)

In addition, we have included two additional HOs, which performance is not as good as HO-10, but that enrich such discussion (see next response and response 1, Rev #3).

3) Their data suggests that the period length of the semi-synthetic loop-based oscillator (HO-10) is longer than WT and peaks slightly earlier than WT (Fig.2). Is this the general property of this "hybrid oscillator" or just specific to HO-10? This reviewer appreciates if authors discuss this point in Discussion.

We interpret that altered phase information is a particular property of the HO-10 (as discussed in the MS). Yet, when it comes to period, our experience indicates that these HOs tend to exhibit periods close to, but longer than WT. And while one of the new HO included in the revised MS (HO-vvd) depicts a short period, it is important to highlight that HO-vvd yields only two peaks, which makes period calculation complicated, as the first peak may be associated to transient effects. This has been now discussed in the text

Line 411: *"In our current limited experience, period of the HOs appears to be longer than WT, and although the two peaks seen in HO-vvd may be suggestive of a shorter period, we cannot discard the effect of transients that may obscure proper period determination. We foresee that detailed study of these and other semi-synthetic circuits will continue to yield important circadian insights"*.

4) The reviewer believes Fig.5E supports simulation data that they showed in (D). The reviewer appreciates if authors quantitatively and statistically analyze data (E) so that readers easily compare D and E.

The experimental result confirms a statistically significant phase advance (~ 3 h), although of a lesser magnitude than the predicted one (8 h).

This is now mentioned in the text

Line 350: *"Our experimental data confirmed that, as predicted in our model, a LD12:12 regime causes a significant phase advance compared to LD4:20 (~ 3 h), albeit of a smaller magnitude than expected"*

5) The circadian rhythm changes many physiological aspects, and environmental factors also affect circadian machinery. Did author observe any physiological change of HO-10 such as the metabolic state and proliferation?

From a phenotypic perspective HO-10 does not exhibit any particular trait

accusative of physiological alterations (i.e. levels of conidiation/ carotenoids). We hope that us/others can analyze the long-term consequences of a lights-on vs lights-off timer logic, a question that we consider fascinating.

6) A part of labels at Fig.4H is out of sight.
Thanks for noticing that. The figure has been fixed.

7) There is no explanation for the asterisk and the error bars in Fig.EV4.
This has been fixed as well

Referee #3:

The authors take on a fascinating question that is fundamental to the field of circadian biology--whether the native and highly conserved architecture of transcriptional regulation is required for rhythmic function. The manuscript reports the creation of a semi-synthetic oscillatory strain HO-10 where *frq* is placed under the control of the *con-10* promoter, known to be a clock-controlled promoter in constant darkness in the wildtype.

The strongest conclusions of the manuscript, in my view, are that the amplitude and entrained phase of the resulting HO-10 oscillator are plastic and depend presumably on specific regulatory elements in the *con-10* promoter. By contrast, the period of oscillation and temperature compensation properties are highly robust to this "rewiring" as nicely shown by the mutational study presented in Fig 2.

1) However, I was left wondering how strong the evidence is that TTFL regulation is absolutely required for free-running rhythms in this system. An alternative hypothesis might be that the *con-10* promoter works not because it is rhythmic but because it gets close to matching the WT expression level of *FREQ* in DD (Fig EV4 suggests that the other semisynthetic HO systems had markedly higher levels of *FREQ*). The clearest evidence that rhythmic expression of *FREQ* is required would be to show that a non-rhythmic promoter with similar strength to *Pcon-10* gives completely arrhythmic output in molecular assays. If the tools do not exist to do this experiment, I could certainly be convinced that it is beyond the scope of this study.

HO-10 presents strong and stable oscillations for several days. In addition, we generated two other HOs with the promoters of *tubulin* (HO-*tub*) and *vivid* (HO-*vvd*) controlling *frq* transcription, data that have been included in the revised MS. Both promoters have rhythmic expression in constant darkness, and when they control *frq* expression weak oscillations are observed. In the case of HO-*tub* oscillations have a small amplitude and tend to dampen, whereas HO-*vvd* only has two peaks before dampening. HO-*tub* has similar levels of *FRQ* in DD compared to HO-10 (~1.1X) and HO-*vvd* has ~0.5X level of *FRQ* compared to HO-10 and ~0.7X compared to WT in DD. These results (Fig EV5) have been included and discussed

Line 168: "We also examined *FRQ* levels in two additional HOs that exhibit oscillations, HO-*tub* and HO-*vvd*, generated with the *tubulin* and *vivid* promoter, confirming that *FRQ* levels in DD were similar (~1.1X) or lower (~0.5X) than the ones in HO-10 (Fig EV5). HO-*tub* and HO-*vvd* were developed based on the promoters of a weakly and strongly rhythmic gene, respectively (Fig EV5A

and B). After LD entrainment HO-*vvd* exhibits only two peaks and then loses rhythmicity, whereas HO-*tub* shows low amplitude oscillations with an unstable period of 24.72 ± 3.06 h (Fig EV5C and D)."

Line 403: "It is also noteworthy that one of the other semi-synthetic oscillators tested in this work (HO-*tub*) exhibited an oscillatory behavior (**Fig EV5**), despite bearing a promoter that is loosely connected to the output pathways, reinforcing the idea of the genetic plasticity of a functional circadian oscillator. In contrast HO-*vvd*, which is based on the *vvd* promoter (a direct target of WCC that displays rhythmic transcription (Cesbron *et al*, 2013)), exhibits limited oscillations, reflecting that it is not all about the wiring of the circuit, but also dependent on the transcriptional landscape and kinetics governing expression of the Negative Element (Oehler *et al*, 2023)"

2) other comments:

* "Temperature compensation is unlikely a network property"--meaning is somewhat unclear. Do the authors specifically mean "transcriptional network"?

Thanks for the opportunity to clarify this. We have now better explained it, including additional references.

Line 225: "Temperature compensation is unlikely a network-wide process"

Line 249: "*the results strongly indicate that transcriptional-wide processes are not playing a key role, as predicted by a network model of temperature compensation, (Kurosawa & Iwasa, 2005), and that instead such clock property is expected to depend mainly on translational/posttranslational mechanisms.*"

Line 423: "*These results further indicate that this emergent property is unlikely to depend on a transcriptional network property, as expected from a network model of temperature compensation (Kidd *et al*, 2015; Kurosawa & Iwasa, 2005)"*

3) *I find figure 5B somewhat confusing. The bifurcation diagram in black seems to imply that the rate constant describing activity of the con-10 promoter is treated as a parameter of the model (on the horizontal axis) and a Hopf bifurcation occurs at a critical value of this rate constant. But then the red arrow describing the effect of a discrete light pulse seems to suggest that the activity of the parameter is permanently changed by the light pulse pushing the system into the oscillatory region. This suggests that the "parameter" k_{24} is actually a dynamical variable in the model that can remember prior stimulation by a light pulse. Please clarify how we should read this figure and how the light pulse treatment is handled in the model.

We have explained this better in the text

Line 314: "Specifically, *con-10* mRNA shows rapid induction followed by photoadaptation when *Neurospora crassa* is transferred from dark to light. Hence, we assumed that the rate of *con-10* expression (k_{24}) undergoes transient light-dependent increase"

Line 324: "*Specifically, a light pulse increases the rate of con-10 expression (k_{24}) pushing the system from a stable steady state to an unstable steady state with a stable periodic limit cycle domain enabling autonomous oscillations (red arrow, in Fig 5B). On the other hand, if HO-10 is grown in LL, then the rate of con-10 expression (k_{24}) is decreased due to the lower activity of con-10 promoter moving the system to a region of stable steady states (blue arrow, Fig 5B)"*

Dear Luis,

Thank you for submitting a revised version of your manuscript. I sincerely apologise for the protracted assessment process due to delays in referee comment submission and the post-holiday backlog.

Your study has now been seen by all original referees. Reviewers #2 and #3 find that their previous concerns have been addressed and now broadly recommend acceptance of the manuscript, while reviewer #1 indicates remaining concerns (below and in the attached file), which I would ask you to address with by extending discussion in the final revision. Please also address the remaining points by reviewer #2.

There now remain only a few editorial points that need addressing before I can extend acceptance of the manuscript:

1. Please submit up to five keywords.
2. Please refer to the figure 2A-G in the manuscript text.
3. We can accommodate up to five EV figures. Please move the rest of the EV figures and their legends to the Appendix, which should be prefaced with a brief table of contents. Further information is available here: <https://www.embopress.org/page/journal/14602075/authorguide#expandedview>.
4. Our data editors have flagged the following issues in figure legends that need correcting:
 - Please note that the legend for figure EV 5-7 has been interchanged. The legend for figure EV 6a-b is provided in figure EV 7 as EV 7a-b. Similarly, the legends for figures EV 7a-f are provided in figure EV 5 as EV 5a-f. And the legends for figure EV 5a-f are provided in figure EV 6 as 6a-f. This needs to be rectified.
 - Please note that the measure of center for the error bars needs to be defined in the legends of figures 4e-f; EV 4c-d.
5. Please upload tables EV1-3 as individual, editable files.
6. Please move the "Data and materials availability" section to the end of "Materials and Methods" and rename it into "Data availability".
7. CRedit has replaced the traditional author contributions section because it offers a systematic, machine-readable author contributions format that allows for more effective research assessment. Please remove the Authors Contributions from the manuscript and use the free text boxes beneath each contributing author's name in our online submission system to add specific details on the author's contribution. More information is available in our guide to authors.
8. In the source data for Fig 1F, the frq mutant panel does not appear to fit to the main figure, please check.
9. Papers published in The EMBO Journal are accompanied online by a 'Synopsis' to enhance discoverability of the manuscript. It consists of A) a short (1-2 sentences) summary of the findings and their significance, B) 3-4 bullet points highlighting key results and C) a synopsis image that is 550x300-600 pixels large (width x height, jpeg or png format). You can either show a model or key data in the synopsis image. Please note that the image size is rather small and that text needs to be readable at the final size. Please send us this information together with the revised manuscript.

With best wishes,

Ieva

We realize that it is difficult to revise to a specific deadline. In the interest of protecting the conceptual advance provided by the work, we recommend a revision within 3 months (28th Apr 2024). Please discuss the revision progress ahead of this time with the editor if you require more time to complete the revisions.

Referee #1:

con-10 promoter control is very different from frq promoter control and accordingly the con-10 promoter fragment inserted in front of the frq ORF changes the quality of transcription control (by altering quality and quantity of TF binding or changing the "TF-mix" binding to the promoter and their contribution to gene control). That much is obvious and very interesting. It is remarkable that HO-10 is rhythmic and that CSP-1 and SUB-1 apparently have a significantly stronger influence on light dependent and circadian expression of frq in this strain.

However, most of the manuscript and its conclusions rely on the assumption that the WCC does not directly bind to and control the hybrid con-10/frq promoter in HO-10. Yet, the absence of hits in deep sequencing data is insufficient proof of that and the available evidence on con-10 transcriptional regulation in the literature (as already stated in my first comments) suggest otherwise. Thus, the revised version of the manuscript still does not convince me that the WCC is not the main essential TF directly controlling frq in HO-10.

In Fig. Reb1 of their response the authors show a graph obtained from the microarray experiments of Chen et al. 2009. They state that con-10 is a late light response gene because transcription peaks later than WCC-controlled frq and vvd. However, vvd reaches its expression peak later than con-10 (after 45 min) and Chen et al. categorize con-10 as an early light responsive gene (see Chen et al. suppl. tables II, IV and VII). Definitions aside, the peak of expression in response to light is not a criterium to distinguish between genes controlled directly (early) and indirectly (late) by the WCC because peak levels also depend on e.g. RNA stability and adaptation dynamics. The only criterium is whether or not the gene responds to light faster than it would take WCC to induce sufficient expression of a 2nd level transcription factor, which in turn activates expression of the late light responsive gene.

Thus, the graph in Fig. Reb1 (see also figure below) supports my argument as it clearly shows that con-10 transcription rapidly reacts to light stimulation by increasing significantly already after 5 minutes of exposure as opposed to e.g. the late light responsive gene NCU01107 (green, data from Chen et al.), which shows a significant delay in light induced expression.

As WCC is the only known light activatable TF in Neurospora, indirect light responsive gene expression on such short time scale is very unlikely, thereby providing convincing evidence that light induction of con-10 transcription is under direct WCC control. In consequence, this would mean that the clock in HO-10 is still a one-step TTFL controlled by the WCC and not completely rewired. Rather, the clock in HO-10 is modulated with some altered interesting properties.

To avoid confusion down the road, I encourage the authors to test and compare timing of response to light pulses in their frq and con-10 luc-PEST reporter strains in the wt background.

Referee #2:

In this study, Goity and co-authors reported a semi-synthetic clock in Neurospora by rewriting the transcriptional circuits. Throughout this review process, the manuscript has been further improved. The following comments are just intended to enhance the clarity of the manuscript.

Minor comments:

1. In the current manuscript, it emphatically asserts that CSP-1 plays a crucial role not only in the phase regulation of WT but also in the noncanonical regulation of the HO-10 system. The authors have added some explanation, but a little more introduction to CSP-1 would help readers outside of chronobiology to understand its background.

2. This reviewer feels that the statement "Future studies will focus on which other TFs, associated to the output pathways, are also essential core-components of this hybrid oscillator circuitry (Goity et al., in preparation)" (Lines 198-200) would be better placed in the discussion section, not in the results section.

3. The figure legends of Fig EV 5-7 are misaligned.

Referee #3:

The authors have addressed all of my concerns--I'm very satisfied with the revised manuscript and think it should be published.

Referee #1:

con-10 promoter control is very different from frq promoter control and accordingly the con-10 promoter fragment inserted in front of the frq ORF changes the quality of transcription control (by altering quality and quantity of TF binding or changing the "TF-mix" binding to the promoter and their contribution to gene control). That much is obvious and very interesting. It is remarkable that HO-10 is rhythmic and that CSP-1 and SUB-1 apparently have a significantly stronger influence on light dependent and circadian expression of frq in this strain.

However, most of the manuscript and its conclusions rely on the assumption that the WCC does not directly bind to and control the hybrid con-10/frq promoter in HO-10. Yet, the absence of hits in deep sequencing data is insufficient proof of that and the available evidence on con-10 transcriptional regulation in the literature (as already stated in my first comments) suggest otherwise. Thus, the revised version of the manuscript still does not convince me that the WCC is not the main essential TF directly controlling frq in HO-10.

In Fig. Reb1 of their response the authors show a graph obtained from the microarray experiments of Chen et al. 2009. They state that con-10 is a late light response gene because transcription peaks later than WCC-controlled frq and vvd. However, vvd reaches its expression peak later than con-10 (after 45 min) and Chen et al. categorize con-10 as an early light responsive gene (see Chen et al. suppl. tables II, IV and VII). Definitions aside, the peak of expression in response to light is not a criterium to distinguish between genes controlled directly (early) and indirectly (late) by the WCC because peak levels also depend on e.g. RNA stability and adaptation dynamics. The only criterium is whether or not the gene responds to light faster than it would take WCC to induce sufficient expression of a 2nd level transcription factor, which in turn activates expression of the late light responsive gene.

Thus, the graph in Fig. Reb1 (see also figure below) supports my argument as it clearly shows that con-10 transcription rapidly reacts to light stimulation by increasing significantly already after 5 minutes of exposure as opposed to e.g. the late light responsive gene NCU01107 (green, data from Chen et al.), which shows a significant delay in light induced expression.

As WCC is the only known light activatable TF in Neurospora, indirect light responsive gene expression on such short time scale is very unlikely, thereby providing convincing evidence that light induction of con-10 transcription is under direct WCC control. In consequence, this would mean that the clock in HO-10 is still a one-step TTFL controlled by the WCC and not completely rewired. Rather, the clock in HO-10 is modulated with some altered interesting properties.

To avoid confusion down the road, I encourage the authors to test and compare timing of response to light pulses in their frq and con-10 luc-PEST reporter strains in the wt background.

We thank the insightful observations. Indeed, since in this work we have not fully discarded (nor shown) WCC direct binding on the *con-10* promoter, we have acknowledged such limitation in the discussion).

"Thus, a limitation of the current study is that part of the resulting circuitry may still relay on a direct effect of WCC, which would mean that the core-clock topology would be an admixture of a classic TTFL circuit, entangled with multiple output nodes." And

"Future efforts will be focused on providing a detailed transcriptional landscape of this rewired oscillator, including the main TFs acting as bona fide positive elements, particularly unravelling the exact direct contribution of WCC."

"Thus, this work uncovers that the evolutionarily conserved simple one-step TTFL is not the only possible functional circadian core-clock topology that could arise based on already existing cellular components."

Referee #2:

In this study, Goity and co-authors reported a semi-synthetic clock in *Neurospora* by rewriting the transcriptional circuits. Throughout this review process, the manuscript has been further improved. The following comments are just intended to enhance the clarity of the manuscript.

Minor comments:

1. In the current manuscript, it emphatically asserts that CSP-1 plays a crucial role not only in the phase regulation of WT but also in the noncanonical regulation of the HO-10 system. The authors have added some explanation, but a little more introduction to CSP-1 would help readers outside of chronobiology to understand its background.

We have added a phrase providing more context

"We chose CSP-1, a TF known to be under clock control and that is part of the output pathways, but that it is not essential for clock function. csp-1 encodes for a regulator that holds similarities to the yeast transcription repressors NRG1 and NRG2, and it has been described to have a large role in regulating metabolic genes in Neurospora (Sancar et al, 2011). Relevantly, it represents one of the best studied TF involved in circadian output, modulating a large number of ccgs, including con-10"

2. This reviewer feels that the statement "Future studies will focus on which other TFs, associated to the output pathways, are also essential core-components of this hybrid oscillator circuitry (Goity et al., in preparation)" (Lines 198-200) would be better placed in the discussion section, not in the results section.

Thanks for pointing this out, this was modified

3. The figure legends of Fig EV 5-7 are misaligned.

Thanks for noticing. This has been now fixed

Referee #3:

The authors have addressed all of my concerns--I'm very satisfied with the revised manuscript and think it should be published.

Thanks!

Dear Luis,

Thank you for addressing the final editorial issues. I am now pleased to inform you that your manuscript has been accepted for publication.

I will look into the synopsis text in the next couple of days and let you know if any edits to the journal style are needed.

If you have any questions, please do not hesitate to contact the Editorial Office. Thank you for this contribution to The EMBO Journal and congratulations on a nice study!

Best wishes,

leva

leva Gailite, PhD
Senior Scientific Editor
The EMBO Journal
Meyerhofstrasse 1
D-69117 Heidelberg
Tel: +4962218891309
i.gailite@embojournal.org
